# A Monte Carlo Framework for Calibrated Uncertainty Estimation in Sequence Prediction

**Qidong Yang**\*                                                                 *qidong@mit.edu*
*Massachusetts Institute of Technology*

**Weicheng Zhu**\*                                                                 *jackzhu@nyu.edu*
*New York University*

**Joseph Keslin**                                                                 *jkeslin2@illinois.edu*
*University of Illinois Urbana-Champaign*

**Laure Zanna**                                                                 *laure.zanna@nyu.edu*
*New York University*

**Tim G. J. Rudner**                                                                 *tim.rudner@utoronto.ca*
*University of Toronto*

**Carlos Fernandez-Granda**                                                                 *cfgranda@cims.nyu.edu*
*New York University*

**Reviewed on OpenReview:** *https://openreview.net/forum?id=sJE59flFC1*

## Abstract

Probabilistic prediction of sequences from images and other high-dimensional data remains a key challenge—particularly in safety-critical domains. In these settings, it is often desirable to quantify the uncertainty associated with a prediction in addition to determining the most likely sequence. In this paper, we consider a Monte Carlo framework to estimate probabilities and confidence intervals associated with sequences. The framework uses a Monte Carlo simulator, implemented as an autoregressively trained neural network, to sample sequences conditioned on an image input. We then use these samples to estimate probabilities and confidence intervals, which are then evaluated using a suite of customized uncertainty evaluation metrics. Experiments on synthetic and real data show that the framework produces accurate discriminative predictions, but can suffer from miscalibration. To address this shortcoming, we propose a time-dependent regularization method, which produces calibrated predictions.

## 1 Introduction

We consider the problem of predicting a sequence of multi-class labels from high-dimensional input data, such as images. Potential applications include patient prognostics from medical imaging data (Pham et al., 2017), weather forecasting (Yang et al., 2025; Peduto et al., 2026), and modeling of human behavior (Thakkar et al., 2024). Our focus is on *probabilistic* prediction, which requires uncertainty quantification in the form of probabilities or confidence intervals.

Figure 1 shows two examples of sequence prediction problems: modeling player behavior in a video game and health monitoring. The data consist of "starting state" images and subsequent sequences of player actions (e.g., *move left*, *move right*, *shoot*, etc.) and patients' health statuses (*healthy*, *ill* or *dead*), respectively. The goal is to predict the sequence of actions/health statuses from the starting state image.

---

\*Equal contribution.

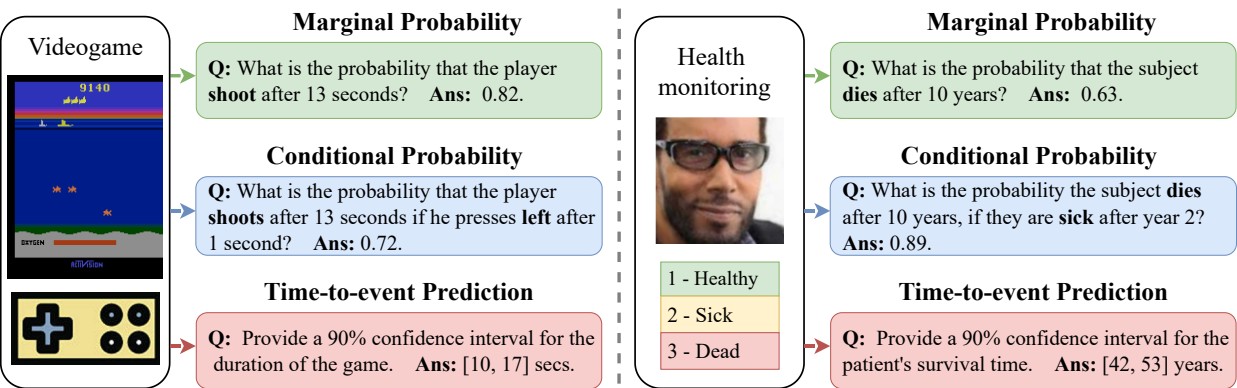

Figure 1: **Sequence prediction with uncertainty estimation.** The proposed framework enables estimation of marginal probabilities, conditional probabilities, and time-to-event confidence intervals associated with a sequence given an input image. We consider sequential decision-making tasks where the input image is a screenshot from an Atari video game and the sequence to predict is the sequence of actions taken by a user (left), and we consider a synthetic-data forecasting problem where the input image is the face of a person and the sequence to predict is the evolution of their health status (right).

There has been a large volume of work on estimating the *most likely* sequence from high-dimensional data, for example, in language modeling (You et al., 2016; Herdade et al., 2019; Li et al., 2022; Brown et al., 2020). However, predicting a most likely sequence is *not sufficient* when the evolution of the sequence is uncertain. In this case, multiple states are possible at a future time given a fixed input, and we need to predict the probability that each state occurs at a given point in the future. Similarly, in prediction of time-to-event values (e.g., the time until the game ends, or the survival time of a subject), predicting the most likely value is insufficient, as there is a range of possible times given a specific input. Instead, we wish to produce confidence intervals that are likely to contain the time of interest with high probability—a prediction task that remains largely underexplored. While predicting the probability of the different possible states of a sequence at a fixed future time is a standard classification and probability-estimation problems (Liu et al., 2022), existing approaches require separate models for each time and type of event. In this paper, we propose a framework that uses a single model to generate probabilities and confidence intervals associated with any possible future time and any type of event.

The proposed **f**ramework for M**o**nte **C**arlo **u**ncertainty quantification of **s**equences (**foCus**) combines autoregressive models—which are the state of the art for many sequence-based tasks (Bahdanau et al., 2014; Gehring et al., 2017; Vaswani et al., 2017; Brown et al., 2020), with Monte Carlo estimation to produce useful uncertainty estimates for sequence prediction tasks. To support systematic evaluation, we also propose a suite of metrics tailored to sequence-derived uncertainty estimates (Section 4.1). However, when studying this framework, we find that autoregressive simulators trained via maximum likelihood estimation are severely miscalibrated, meaning that the associated probabilities or confidence intervals do not provide accurate uncertainty quantification for the tasks at hand. To address this shortcoming, we develop a time-dependent logit regularizer for autoregressive simulator training and show that it enables foCus to generate better calibrated probability estimates.

To summarize, our main contributions are as follows:

1. We propose a Monte Carlo framework to study an under-explored task, probabilistic sequence prediction, with high-dimensional input data using autoregressive models.

2. We propose a suite of evaluation metrics tailored to sequence-derived uncertainty estimates.

3. We perform an empirical evaluation of the proposed framework on a hand-tailored synthetic benchmarking task for sequence prediction and on non-synthetic sequential decision-making tasks and find that neural network-based autoregressive simulators are prone to time-dependent miscalibration.

4. We develop a time-dependent regularization method and show that it allows learning simulators that produce calibrated uncertainty estimates under the proposed framework.

> Code to reproduce our experiments can be found at https://github.com/qy707/Deep_Probability.

## 2 Related Work

**Sequence generation.** Discrete sequence generation plays a fundamental role in many natural language processing applications, such as language modeling (Brown et al., 2020; Touvron et al., 2023), image captioning (Ghandi et al., 2023), language translation (Bahdanau et al., 2014; Gehring et al., 2017; Vaswani et al., 2017), and text summarization (Dong, 2018). In these examples, sequence generation is typically performed in an autoregressive manner, where each token is generated based on the tokens previously generated. Alternatively, sequences can also be generated non-autoregressively (Sun & Yang, 2020; Gu et al., 2017; Shu et al., 2020), where tokens are produced simultaneously or with fewer dependencies on earlier tokens. In both paradigms, the primary objective is to generate the most likely sequence. However, these approaches typically do not focus on quantifying the uncertainty associated with generated sequences, which is the goal of this paper.

**Imitation and reinforcement learning.** Our focus is on *predicting* sequences from high-dimensional data, which is fundamentally different from imitation learning (Hussein et al., 2017), which seeks to replicate human behavior, and reinforcement learning (Sutton, 2018), which seeks to determine an optimal policy by allowing the agent to interact with the environment, guided by a reward function (Schulman et al., 2017; Lillicrap et al., 2019).

**Calibration.** Miscalibration is a well-known challenge in classification models based on deep learning (Guo et al., 2017; Wang, 2024; Wang et al., 2021), particularly when the goal is to provide accurate uncertainty quantification (Liu et al., 2022). While significant progress has been made in calibrating classification models, calibration in sequence prediction tasks remains underexplored and even lacks a clear definition. While Marx et al. (2024) explores calibration in sequences, their focus is on step-wise calibration, where each sequence step has an input-output pair, which makes their setting step-wise calibration within sequences, whereas our work focuses on calibration for the entire sequence. Kuleshov & Liang (2015) offers a framework for measuring calibration in structured high-dimensional random vectors via event pooling, which inspires our approach to uncertainty estimation in sequence prediction. Various methods have been developed to enhance calibration in classification, including post-processing the logits (Gupta et al., 2021; Kull et al., 2017; 2019), ensembling methods (Zhang et al., 2020a; Lakshminarayanan et al., 2017; Maddox et al., 2019), soft labeling (Mukhoti et al., 2020; Szegedy et al., 2016; Zhang et al., 2017a; Thulasidasan et al., 2019; Liu et al., 2022) and training with regularization (Pereyra et al., 2017; Kumar et al., 2018; Rudner et al., 2023). In this work, we propose a time-dependent regularization method to improve calibration specifically in sequence prediction tasks.

**The Monte Carlo method in Bayesian deep learning.** In Bayesian deep learning, the Monte Carlo method is often used to generate a Bayesian model average using samples drawn from a distribution over parameters (Blundell et al., 2015; Gal & Ghahramani, 2016). In contrast, in our proposed framework we sample sequences using a neural-network simulator with fixed parameters, and apply the Monte Carlo method to the sampled sequences. Monte Carlo methods are also used for uncertainty estimation on large language model outputs (Malinin & Gales, 2021; Jiang et al., 2021; Kuhn et al., 2023; Xiong et al., 2024), where multiple sequences are generated to assess the confidence of factual outputs and mitigate hallucinations. Unlike these works, our goal is not to determine the most likely sequence, but rather to provide a probabilistically-accurate characterization of possible future sequences.

## 3 A Monte Carlo Framework for Uncertainty Estimation in Sequence Prediction

We consider the problem of predicting a sequence of multi-class labels from high-dimensional data. More formally, our goal is to estimate the conditional distribution of a sequence $Y$ consisting of $\ell$ discrete random variables each with $c$ possible states given an observed input $x$, interpreted as a sample of a random vector $X$. In our health-monitoring example, the states are *healthy*, *ill*, and *dead*, and $X$ represents an input image. The framework extends naturally to any type of modalities.

We focus on discrete sequences because this setting is already high-dimensional, with a combinatorially exploding space of outcomes, and because it is relevant to safety-critical applications such as clinical-state trajectories and discrete action sequences. A thorough study of this setting is a key step towards extending the framework to continuous-valued or function-valued settings.

Even for short sequence lengths, directly estimating the joint conditional probability mass function of $Y$ given $X = x$ is intractable due to the combinatorial explosion of possible sequences (e.g., for $c := 3$ and $\ell := 100$ there are $3^{100} > 10^{47}$ possible sequences!), which is an instance of the notorious curse of dimensionality. Instead, we propose to estimate the following probabilities and confidence intervals characterizing the sequence, which are illustrated in Figure 1:

1. The *marginal* probability $\mathrm{P}(Y_i = a \mid X = x)$ that the $i$th entry $Y_i$ of the sequence is equal to $a \in \{1, ..., c\}$. We refer to this as a marginal probability, but strictly speaking it is a conditional probability, as it is conditioned on $X = x$. In health monitoring, this is the probability that a subject is healthy, ill, or dead at time $i$.

2. The *conditional* probability $\mathrm{P}(Y_i = a \mid Y_j = b, X = x)$ that the $i$th entry $Y_i$ of the sequence is equal to $a \in \{1, ..., c\}$ given that the $j$th entry is equal to $b \in \{1, ..., c\}$. In health monitoring, this could be the conditional probability that a subject is dead at time $i$ given that they are ill at time $j$.

3. The $\alpha$ *confidence interval* $I_\alpha$ for the time $\widetilde{T}$ until a certain event associated with the sequence occurs (e.g., $\widetilde{T} := \min\{i : Y_i = a\}$ for some $a \in \{1, ..., c\}$), which should satisfy $\mathrm{P}(\widetilde{T} \in I_\alpha) = \alpha$, where $\alpha$ is typically set to 0.9 or 0.95. In health monitoring, $\widetilde{T}$ can represent the time until death or recovery.

### 3.1 Monte Carlo Estimation

As noted in the previous section, a key challenge in estimating arbitrary probabilities and confidence intervals associated with a sequence of random variables $Y$ is that it is intractable to *explicitly* estimate the joint distribution given the input $X$ (unless we make highly simplifying modeling assumptions, such that the sequence forms a Markov chain). Our proposed **f**ramework for M**o**nte **C**arlo **u**ncertainty quantification of **s**equences (**foCus**) addresses this challenge by instead *implicitly sampling* from the conditional distribution using a neural network simulator, described in Section 3.2.

Given an input $x$, we apply the simulator to generate $M$ sequences $\{(\hat{y}_1^{(m)}, ..., \hat{y}_l^{(m)})\}_{m=1}^M$. As shown in panel (c) of Figure 2, the sequences are then used to estimate any desired probability or confidence interval. The marginal probability of state $a$ at time $i$ is estimated by the fraction of sequences in state $a$ at time $i$:

$$\mathrm{P}(Y_i = a \mid X = x) = \frac{1}{M} \sum_{m=1}^M \mathbb{1}_{\{\hat{y}_i^{(m)}=a\}}. \tag{1}$$

The conditional probability of state $a$ at time $i$ given state $b$ at time $j$ is estimated by the fraction of sequences in state $a$ at time $i$ out of sequences in state $b$ at time $j$:

$$\mathrm{P}(Y_i = a \mid Y_j = b, X = x) = \frac{\sum_{m=1}^M \mathbb{1}_{\{\hat{y}_i^{(m)}=a, \hat{y}_j^{(m)}=b\}}}{\sum_{m=1}^M \mathbb{1}_{\{\hat{y}_j^{(m)}=b\}}}. \tag{2}$$

To estimate the $\alpha$ confidence interval $I_\alpha$ of the time-to-event $\widetilde{T}$, we first compute the value of $\widetilde{T}$ associated with each simulated sequence, which yields $M$ times, $\{T^{(1)}, ..., T^{(M)}\}$. These times are sorted to calculate the $(1 - \alpha)/2$ and $(1 + \alpha)/2$ percentiles $q_{(1-\alpha)/2}$ and $q_{(1+\alpha)/2}$ that are used to build the confidence interval $I_\alpha = [q_{(1-\alpha)/2}, q_{(1+\alpha)/2}]$.

### 3.2 Autoregressive Simulation

In order to produce the simulated sequences required by our Monte Carlo framework, we employ a neural-network simulator. The simulator consists of a convolutional neural network (CNN) that encodes the input image $x$, and a recurrent neural network (RNN) that iteratively estimates the conditional distribution of the $i$th entry $Y_i$ of the sequence given $X = x$ and the values of all previous entries, i.e. $\mathrm{P}(Y_i = y_i \mid X = x, Y_1 = y_1, ..., Y_{i-1} = y_{i-1})$.

Figure 2(b) illustrates how the simulator is used to obtain a sample sequence $(\hat{y}_1, \ldots, \hat{y}_\ell)$. The input image $x$ is fed into the CNN, producing a hidden vector $h_0$ that is fed into the RNN to then generate the simulated sequence iteratively. At each iteration $i \in \{1, ..., \ell\}$, the input of the RNN is the value $\hat{y}_{i-1}$ of the previous entry (except for $i = 1$) and the hidden vector $h_{i-1}$. The outputs are an estimate of the conditional

(a) Training of the autoregressive simulator    (b) Sampling from the simulator given an input image

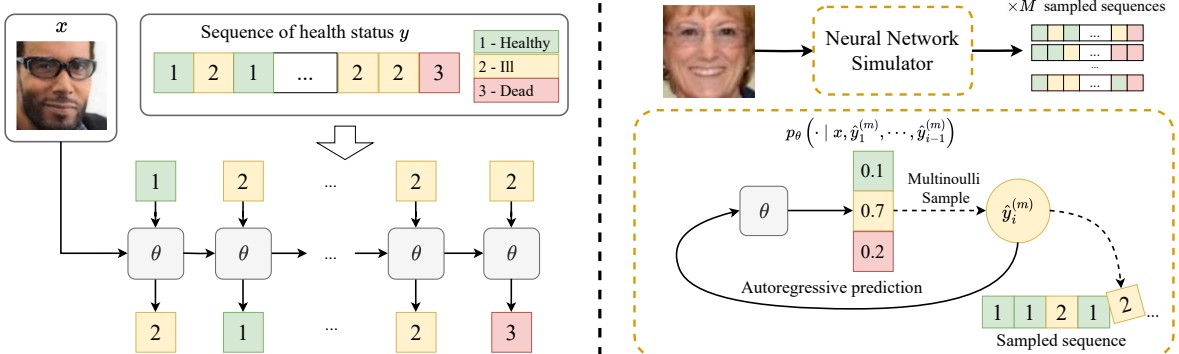

(c) Probabilistic prediction via Monte Carlo estimation

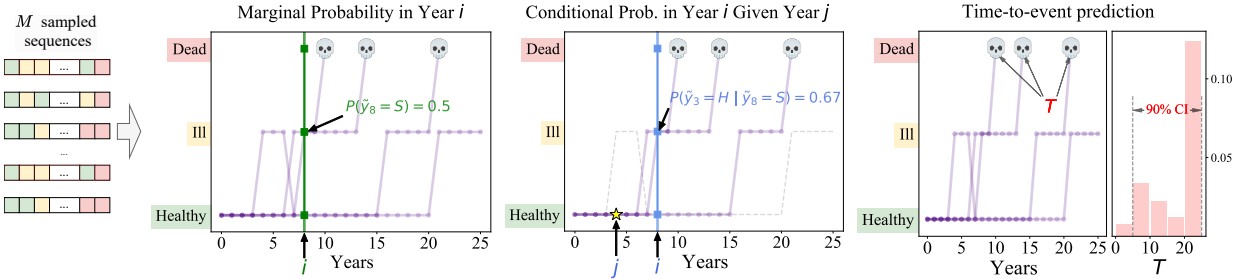

Figure 2: **Monte Carlo framework for uncertainty estimation in sequence prediction.** (a) A neural network simulator is trained to autoregressively predict the conditional distribution of each entry in a sequence given an input image and the preceding states. (b) The simulator is used to generate multiple sample sequences, by iteratively sampling from the estimated conditional distribution. (c) The Monte Carlo method is applied to estimate marginal probabilities, conditional probabilities, and time-to-event confidence intervals from the samples.

distribution of $Y_i$ given the previous entries and $X = x$, and an updated hidden vector $h_i$. The $i$th entry $\hat{y}_i$ of the sample sequence is sampled from this conditional distribution.

The simulator is trained using a dataset of image-sequence pairs, as illustrated in Figure 2(a). During training, the model uses the ground truth value $y_{i-1}$ of the previous entry along with the hidden vector $h_{i-1}$ to predict the subsequent $i$th entry $Y_i$. As explained in Sections 5 and 6 the design of the training loss plays a crucial role in avoiding miscalibration in the downstream probabilities and confidence intervals computed using the simulator.

## 4 Evaluating Uncertainty

### 4.1 Evaluation metrics

We assess marginal and conditional probability estimates with a set of complementary metrics. Macro Area Under the ROC Curve (AUC) quantifies discriminative ability. Expected Calibration Error (ECE) quantifies calibration. Brier Score (BS) and Cross Entropy (CE) provide a more holistic evaluation. These metrics are computed for each entry and then aggregated via averaging to obtain sequence-level metrics.

To assess confidence intervals, we evaluate discriminability via the Mean Absolute Error (MAE) (normalized by the average sequence length). Calibration is evaluated by computing the coverage probability of the intervals, which should be close to the target confidence level $\alpha$. We also measure the average confidence interval width relative to the average sequence length. Further details are provided in Appendix A.

Table 1: **Marginal probability estimation.** The table reports sequence-level metrics evaluating the performance of the proposed foCus framework for estimation of marginal probabilities (see Section 3). We compare versions of foCus without regularization (see Section 5) and with constant and time-dependent regularization (see Section 6). Results are presented as mean ± standard error from three independent model realizations. Time-dependent regularization improves calibration substantially (lower ECE), while maintaining a comparable AUC, which results in superior probability estimates (lower cross entropy and Brier score). Similar results are obtained for conditional probability estimation, as reported in Table 2.

| Scenario | Regularization | ECE (↓) | AUC (↑) | CE (↓) | BS (↓) |
|---|---|---|---|---|---|
| Seaquest | ✗ | $0.0435 \pm 0.0004$ | $0.8671 \pm 0.0035$ | $1.0577 \pm 0.0241$ | $0.1247 \pm 0.0012$ |
|  | time-dependent | $\mathbf{0.0277 \pm 0.0023}$ | $\mathbf{0.8678 \pm 0.0028}$ | $\mathbf{0.6705 \pm 0.0285}$ | $\mathbf{0.1144 \pm 0.0007}$ |
|  | constant | $0.0365 \pm 0.0002$ | $0.8625 \pm 0.0020$ | $0.8173 \pm 0.0068$ | $0.1177 \pm 0.0008$ |
| River Raid | ✗ | $0.0583 \pm 0.0016$ | $\mathbf{0.6453 \pm 0.0009}$ | $1.2034 \pm 0.0281$ | $0.1750 \pm 0.0016$ |
|  | time-dependent | $\mathbf{0.0388 \pm 0.0013}$ | $0.6346 \pm 0.0035$ | $\mathbf{0.8585 \pm 0.0132}$ | $\mathbf{0.1671 \pm 0.0012}$ |
|  | constant | $0.0474 \pm 0.0004$ | $0.6280 \pm 0.0020$ | $1.0274 \pm 0.0158$ | $0.1686 \pm 0.0005$ |
| Bank Heist | ✗ | $0.0559 \pm 0.0032$ | $\mathbf{0.6938 \pm 0.0028}$ | $1.1874 \pm 0.0540$ | $0.2340 \pm 0.0020$ |
|  | time-dependent | $\mathbf{0.0148 \pm 0.0014}$ | $0.6782 \pm 0.0016$ | $\mathbf{0.7647 \pm 0.0130}$ | $\mathbf{0.2166 \pm 0.0005}$ |
|  | constant | $0.0399 \pm 0.0016$ | $0.6928 \pm 0.0046$ | $0.8894 \pm 0.0112$ | $0.2211 \pm 0.0007$ |
| H.E.R.O. | ✗ | $0.0947 \pm 0.0014$ | $0.6785 \pm 0.0061$ | $1.1310 \pm 0.0225$ | $0.1261 \pm 0.0009$ |
|  | time-dependent | $0.0481 \pm 0.0034$ | $\mathbf{0.7159 \pm 0.0105}$ | $\mathbf{0.6940 \pm 0.0246}$ | $\mathbf{0.1170 \pm 0.0008}$ |
|  | constant | $\mathbf{0.0352 \pm 0.0001}$ | $0.7041 \pm 0.0102$ | $0.7218 \pm 0.0391$ | $0.1212 \pm 0.0011$ |
| Road Runner | ✗ | $0.0779 \pm 0.0035$ | $\mathbf{0.6913 \pm 0.0100}$ | $1.1586 \pm 0.0291$ | $0.1575 \pm 0.0034$ |
|  | time-dependent | $\mathbf{0.0204 \pm 0.0012}$ | $0.6823 \pm 0.0084$ | $\mathbf{0.5255 \pm 0.0077}$ | $\mathbf{0.1382 \pm 0.0003}$ |
|  | constant | $0.0303 \pm 0.0027$ | $0.6898 \pm 0.0140$ | $0.6275 \pm 0.0250$ | $0.1394 \pm 0.0010$ |
| FaceMed | ✗ | $0.1503 \pm 0.0048$ | $0.7534 \pm 0.0079$ | $1.6932 \pm 0.0188$ | $0.3464 \pm 0.0012$ |
|  | time-dependent | $\mathbf{0.0757 \pm 0.0068}$ | $\mathbf{0.7614 \pm 0.0024}$ | $\mathbf{0.9085 \pm 0.0303}$ | $\mathbf{0.3328 \pm 0.0008}$ |
|  | constant | $0.0974 \pm 0.0045$ | $0.7613 \pm 0.0028$ | $1.0071 \pm 0.0499$ | $0.3356 \pm 0.0030$ |

## 4.2  Datasets

**FaceMed** is a synthetic dataset designed to predict individual health-status trajectories based on UTKFace (Zhang et al., 2017b), which contains face images of subjects of different ages. For each image, we simulate a sequence of health states using a Markov chain model that depends on the age of the subject (see Appendix B). The health states are *healthy*, *ill*, and *dead*. The goal is to predict the future marginal and conditional health status distributions as well as confidence intervals for subject survival.

**Atari games** is a non-synthetic benchmark created from five human gameplay datasets (Atari-HEAD; Zhang et al., 2020b) consisting of screenshots associated with sequences of player actions. There are 19 actions, including *move left*, *fire*, and an *end* action, which indicates the end of a game sequence. The benchmark includes games:

- Seaquest: Control a submarine to rescue divers while shooting sharks and enemy submarines;
- River Raid: Navigate a fighter jet to destroy enemy targets while managing fuel;
- Bank Heist: Drive through a maze-like city to rob banks while avoiding police;
- H.E.R.O.: Traverse a mineshaft to rescue trapped miners while avoiding enemies and hazards;
- Road Runner: Guide a bird to collect seeds while evading a chasing coyote and obstacles.

The goal is to estimate the marginal and conditional probabilities of player actions, and confidence intervals of the time until the player scores. All datasets follow a strict 7:2:1 train/validation/test split, with test sets held out until final evaluation. Hyperparameters are selected on validation ECE. Each scenario is run for three independent model realizations; we report mean and standard error. For the synthetic FaceMed dataset, we additionally validate against ground-truth marginal probabilities derived from the data-generating process, providing an independent check on model quality. Additional dataset details can be found in Appendix B.

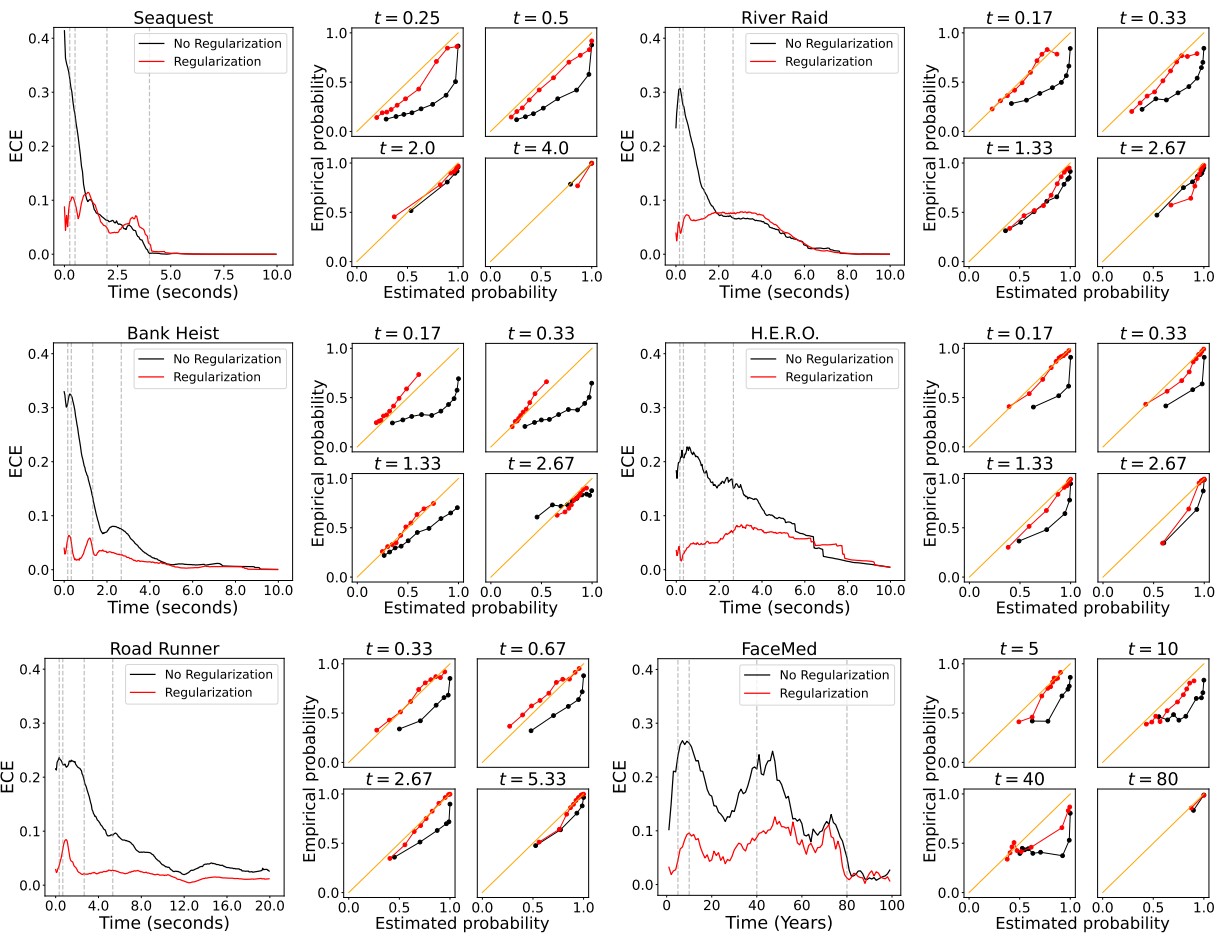

Figure 3: **Entry-wise calibration error and reliability diagrams for marginal probability estimation.** The large graphs plot the entry-level ECE of the proposed foCus framework for estimation of marginal probabilities (see Section 3) without regularization (black line, see Section 5) and time-dependent regularization (red line, see Section 6). Unregularized foCus produces miscalibrated estimates, particularly in the earlier entries, which are dramatically improved by time-dependent regularization for all datasets. The small graph show reliability diagrams for some of the steps, which confirm the improvement in calibration. Additional reliability diagrams and results for constant regularization are shown in Appendix D.1.

## 5 Miscalibration

In this section, we report the results of applying foCus when we train the simulator described in Section 3.2 using a standard unregularized cross-entropy loss

$$\mathrm{CE}(\theta) = \mathop{\mathbb{E}}_{(x,y)\sim\mathcal{D}} \left[ -\sum_{i=1}^{\ell} \log p_\theta \left( y_i \mid x, y_1, ..., y_{i-1} \right) \right], \tag{3}$$

where $x$ is an input image, $y$ is the corresponding sequence and $\mathcal{D}$ is the training set of image-sequence pairs. Here $\theta$ represents the parameters in the neural-network simulator and $p_\theta \left( y_i \mid x, y_1, ..., y_{i-1} \right)$ denotes the corresponding estimate of the conditional probability $\mathrm{P} \left( Y_i = y_i \mid X = x, Y_1 = y_1, ..., Y_{i-1} = y_{i-1} \right)$. Further details about the training procedure are provided in Appendix C.2.

Tables 1 and 2 show that this version of foCus exhibits strong discriminative performance across all our datasets when estimating marginal and conditional probabilities, respectively, indicated by the high sequence-level AUC values. However, the sequence-level ECE values are also high, suggesting that the probability

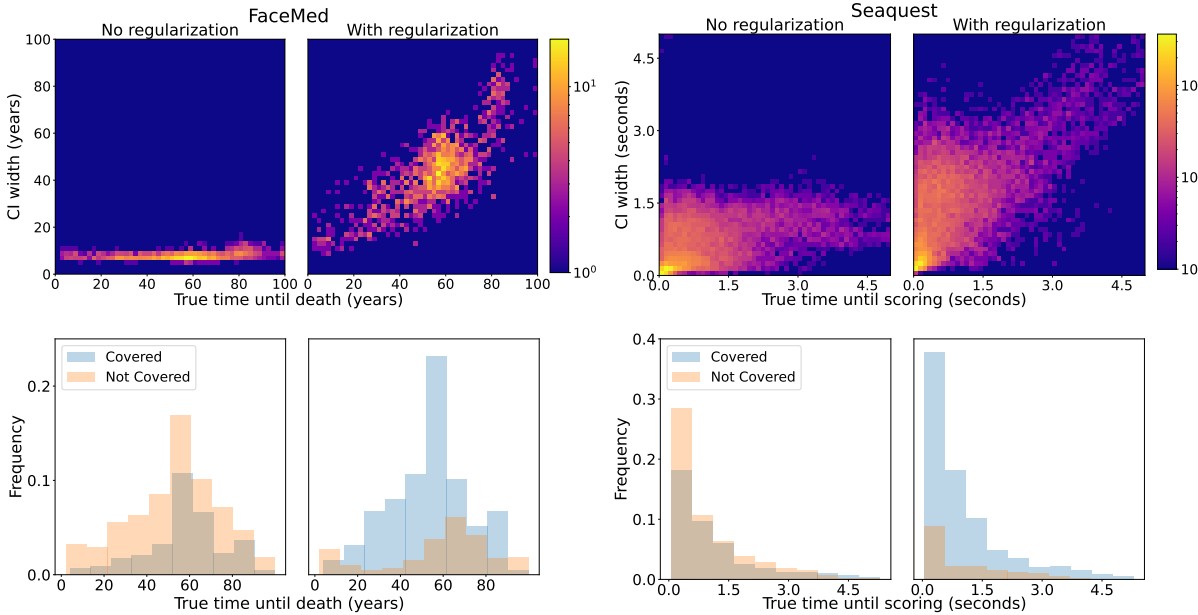

Figure 4: **Confidence intervals for time-to-event prediction and coverage probability.** The upper panel shows heatmaps of the length of 0.9 confidence intervals for time-to-event prediction using the proposed foCus framework without and with time-dependent regularization. The histograms below show the frequency of intervals containing the true times, as a function of the true time. Unregularized foCus produces short intervals with poor coverage, whereas regularization yields intervals that tend to be larger when the ground-truth times are larger, and are much better calibrated. The plots correspond to the FaceMed (left) and Seaquest (right) datasets. Appendix D.3 shows analogous plots for additional datasets.

estimates are not well calibrated. This is corroborated by Figure 3, which shows entry-level ECE values for all datasets. Miscalibration is time-dependent and particularly severe at the beginning of the sequence. The reliability diagrams in Figure 3 show that the model suffers from *overconfidence* (the estimated probabilities are more extreme than the empirical probabilities), which is typical of deep learning models, as they tend to overfit the training labels (Liu et al., 2022).

We also observe miscalibration in the confidence intervals estimated by this version of foCus. Table 3 reports the coverage probabilities computed on test data for $\alpha = 0.9$, which are below 0.5 in all cases except one! Moreover, Figure 4 reveals that the widths of the confidence intervals remain invariant over time, which is undesirable as uncertainty should increase with longer time horizons.

## 6 Time-dependent Logit Regularization for Calibrated Uncertainty Estimation

Calibration requires that predicted probabilities match empirical probability when pooling across events (Kuleshov & Liang, 2015). Neural networks trained via maximum likelihood estimation (MLE) systematically violate this property due to overfitting and memorization: MLE minimizes the cross-entropy between predicted probabilities and one-hot labels, which incentivizes the model to push predicted probabilities toward 0 or 1. In over-parameterized neural networks, this leads to large logit magnitudes that produce overly sharp, low-entropy predictive distributions even when the true target distributions are more uncertain (Guo et al., 2017). Intuitively, the $\ell_2$-norm regularization on logits directly counteracts this mechanism—extreme probabilities near 0 or 1 logits correspond to large $\ell_2$-norm, so penalizing the logit norm prevents overconfident predictions. Through the function-space lens of Rudner et al. (2023), the $\ell_2$-norm regularization corresponds to imposing a Gaussian prior centered at zero in logit space, encoding the belief that extreme probabilities require strong evidence. Under this prior, maximum a posteriori (MAP) estimation shrinks predicted probabilities toward higher-entropy distributions when evidence is insufficient (Qiu et al., 2023; Rudner et al., 2024; 2025; Lopez et al., 2026).

Simply put, we can think of the $\ell_2$-norm penalty as promoting neural network parameters that induce predictive functions with higher predictive entropy. Because this regularizer operates on the output logits rather than internal architecture, it is architecture-agnostic; in Section 6.1, we demonstrate that calibration improvement occurs for both RNN and Transformer simulators. As in Section 5, let $p_\theta(y_i|x, y_1, ..., y_{i-1})$ denote the estimate of the conditional probability $P(Y_i = y_i \mid X = x, Y_1 = y_1, ..., Y_{i-1} = y_{i-1})$ produced by the simulator for $i \in \{1, ..., \ell\}$, which is obtained by feeding a logit vector $\mathbf{z}(x, y_1, ..., y_{i-1}) \in \mathbb{R}^c$ into a softmax function. The training loss is

$$\mathcal{L}(\theta) \doteq \underset{(x,y)\sim\mathcal{D}}{\mathbb{E}} \left[ \sum_{i=1}^{\ell} -\log p_\theta(y_i|x, y_1, ..., y_{i-1}) + \lambda_i \|\mathbf{z}(x, y_1, ..., y_{i-1})\|_2 \right], \tag{4}$$

where $\lambda_i$ is a regularization coefficient that governs the regularization strength when predicting the $i$th entry of the sequence. The regularization $\lambda_i$ is designed to be *time dependent*, motivated by our observation that the baseline version of foCus suffers from different degrees of miscalibration at different entries. Because the regularizer operates on the output logits rather than internal architecture, it is architecture-agnostic.

A crucial challenge is how to select the value of this hyperparameter, given the large dimensionality of the hyperparameter space. We propose a selection procedure, based on the observation that miscalibration in the initial entries is propagated by the autoregressive structure of the simulator (see Section 3.2). Consequently, optimizing the regularization parameter at the beginning of the sequence has more impact on the overall calibration performance of foCus (see Appendix E for additional analysis).

The procedure is as follows:

1. For $1 \leq i \leq k_1$ (where $k_1$ is a hyperparameter) we use the sequence-level ECE of marginal probabilities (see Section 4.1) computed over validation set to iteratively select $\lambda_i$, setting $\lambda_j = 0$ for all $j > i$.

2. For $k_1 < i \leq k_2$ (where $k_2$ is a hyperparameter) we constrain all the parameters to equal the same constant, $\lambda_i = \lambda_{\text{all}}$, selected also based on the validation ECE.

3. For $i > k_2$ we set $\lambda_i = 0$.

We set $k_1 = 3$ and performed hyperparameter optimization to select $k_2$, which typically resulted in small values (see Table 5). Details about the optimal regularization parameters and a comprehensive overview of the hyperparameter search process are provided in Appendix C.4.

## 6.1 Results

**Marginal and conditional probabilities.** Table 1 compares the sequence-level evaluation metrics for marginal probability estimation of foCus with (1) no regularization as described in Section 5, (2) our proposed time-dependent regularization described in Section 6, and (3) constant regularization where the regularization parameter in Equation (4) is set to a single constant $\lambda_i = \lambda_{\text{const}}$ (determined based on validation ECE). All methods achieve similar AUCs in each dataset, indicating a similar discriminative ability. In contrast, the ECE is significantly lower for time-dependent regularization for all datasets, indicating better calibration performance. This results in better probability estimates, as evidenced by the lower cross entropy (CE) and Brier scores (BS). For FaceMed this is confirmed by comparing the estimated probabilities to the ground-truth marginal probabilities. The unregularized baseline method and constant regularization yield RMSEs of $0.1847 \pm 0.0014$ and $0.1754 \pm 0.0028$, respectively, while time-dependent regularization reduces the RMSE to $0.1720 \pm 0.0009$ (see Appendix D.1). The same holds for conditional probability predictions, as reported in Table 2: Time-dependent regularization again significantly improves calibration, and as a result the overall probability estimates. The same calibration improvement holds when the RNN simulator is replaced by a Transformer decoder on Seaquest and FaceMed, demonstrated in Appendix G. Appendix F provides a detailed description of the conditional probability estimates for one of the video games.

Figure 3 further demonstrates the improvement in calibration due to time-dependent regularization. Interestingly, we observe a *calibration propagation* phenomenon, where regularizing a small number of early entries produces improved calibration across the whole sequence. For example, for H.E.R.O., regularization is applied to the first 6 entries (0.2 seconds), yet the ECE improvement is evident up until entry 150 (5 seconds).

Table 2: **Conditional probability estimation.** This table reports sequence-level metrics evaluating the performance of the proposed foCus framework for estimation of conditional probabilities (see Section 3). We compare versions of foCus without regularization (see Section 5) and with constant and time-dependent regularization (see Section 6). Results are presented as mean ± standard error from three independent model realizations. Time-dependent regularization improves calibration substantially (lower ECE), while maintaining a comparable AUC, which results in superior probability estimates (lower CE and BS).

| Scenario | Regularization | ECE (↓) | AUC (↑) | CE (↓) | BS (↓) |
|---|---|---|---|---|---|
| Seaquest | ✗ | 0.0590 ± 0.0035 | 0.8887 ± 0.0045 | 0.9032 ± 0.0779 | 0.1386 ± 0.0016 |
| | time-dependent | 0.0316 ± 0.0017 | **0.8938 ± 0.0009** | **0.6070 ± 0.0110** | 0.1301 ± 0.0053 |
| | constant | **0.0298 ± 0.0017** | 0.8827 ± 0.0021 | 0.6227 ± 0.0278 | **0.1289 ± 0.0007** |
| River Raid | ✗ | 0.0841 ± 0.0008 | **0.7007 ± 0.0046** | 1.4968 ± 0.0497 | 0.2264 ± 0.0031 |
| | time-dependent | **0.0652 ± 0.0012** | 0.6997 ± 0.0036 | **1.1587 ± 0.0117** | 0.2248 ± 0.0018 |
| | constant | 0.0689 ± 0.0005 | 0.6926 ± 0.0026 | 1.3235 ± 0.0434 | **0.2225 ± 0.0015** |
| Bank Heist | ✗ | 0.0534 ± 0.0035 | **0.7254 ± 0.0036** | 1.0618 ± 0.0563 | 0.2287 ± 0.0030 |
| | time-dependent | **0.0184 ± 0.0005** | 0.7033 ± 0.0021 | **0.7616 ± 0.0046** | **0.2141 ± 0.0008** |
| | constant | 0.0405 ± 0.0022 | 0.7149 ± 0.0124 | 0.8431 ± 0.0080 | 0.2168 ± 0.0013 |
| H.E.R.O. | ✗ | 0.0993 ± 0.0039 | 0.7063 ± 0.0002 | 1.0290 ± 0.0371 | 0.1195 ± 0.0021 |
| | time-dependent | 0.0645 ± 0.0087 | **0.7420 ± 0.0132** | **0.7189 ± 0.0289** | **0.1151 ± 0.0029** |
| | constant | **0.0418 ± 0.0019** | 0.7249 ± 0.0141 | 0.7283 ± 0.0572 | 0.1208 ± 0.0024 |
| Road Runner | ✗ | 0.0870 ± 0.0061 | 0.6790 ± 0.0091 | 1.2813 ± 0.0244 | 0.1693 ± 0.0039 |
| | time-dependent | **0.0207 ± 0.0012** | 0.6772 ± 0.0107 | **0.5489 ± 0.0089** | **0.1449 ± 0.0003** |
| | constant | 0.0321 ± 0.0032 | **0.6905 ± 0.0135** | 0.6654 ± 0.0304 | 0.1472 ± 0.0012 |
| FaceMed | ✗ | 0.1488 ± 0.0026 | 0.7533 ± 0.0066 | 1.6428 ± 0.0147 | 0.3465 ± 0.0012 |
| | time-dependent | **0.0888 ± 0.0065** | 0.7603 ± 0.0040 | **1.0224 ± 0.0348** | **0.3350 ± 0.0018** |
| | constant | 0.0984 ± 0.0042 | **0.7652 ± 0.0050** | 0.9875 ± 0.0372 | 0.3354 ± 0.0016 |

**Time-to-event confidence intervals.** Table 3 compares the evaluation metrics for the confidence intervals produced by foCus, again with (1) no regularization, (2) time-dependent regularization, and (3) constant regularization. In this case, we observe a certain trade-off between discriminative performance, quantified by the relative MAE, and calibration, quantified by coverage probabilities. The MAE for models trained without or with constant regularization is consistently lower than those of time-dependent regularization, but the coverage probabilities of time-dependent regularization are a lot closer to 90% (between 69% and 92%, compared to at most 70% for the other two methods).

Figure 4 shows heatmaps of the confidence-interval widths for different ground-truth times (upper panel) for FaceMed and Seaquest (see Appendix H for additional plots), as well as a histogram with the fraction of intervals containing the ground-truth times. We observe that with unregularized simulator training, foCus produces narrow confidence intervals with poor coverage across the board, whereas time-dependent regularization yields intervals that tend to be larger when the true times are larger, achieving much better coverage.

## 7 Discussion and Limitations

In this paper, we studied an important, yet underexplored topic: how to achieve reliable uncertainty quantification when predicting sequences from high-dimensional data. We proposed a Monte Carlo framework based on learned autoregressive simulators that enables flexible estimation of probabilities and confidence intervals. Our experiments on sequential decision-making tasks revealed that simulator models learned via maximum likelihood estimation can lead to severely miscalibrated uncertainty estimates. We showed that this shortcoming can be addressed by training the autoregressive simulator model using a time-dependent regularizer, which we find consistently leads to well-calibrated uncertainty estimates. We show that the time-dependent regularizer is effective on both the RNN simulators discussed in the main text as well as transformer simulators, which we discuss in Appendix G.

Table 3: **Performance comparison of time-to-event prediction confidence intervals (CI) across six scenarios.** The table presents metrics from the same experiments as in Table 1. The time-dependent regularization model achieves significantly better 90% CI ($I_{0.9}$) coverage, indicating improved calibration of uncertainty estimation.

| Scenario | Regularization | Coverage Prob. of $I_{0.9}$ | Relative Width of $I_{0.9}$ | Relative MAE ($\downarrow$) |
|---|---|---|---|---|
| Seaquest | ✗ | $0.3420 \pm 0.0177$ | $0.7401 \pm 0.0149$ | $0.4622 \pm 0.0112$ |
| | time-dependent | $\mathbf{0.7065 \pm 0.0121}$ | $2.1718 \pm 0.0521$ | $0.5297 \pm 0.0079$ |
| | constant | $0.4440 \pm 0.0162$ | $0.9409 \pm 0.0266$ | $\mathbf{0.4538 \pm 0.0069}$ |
| River Raid | ✗ | $0.4463 \pm 0.0105$ | $1.0734 \pm 0.0398$ | $0.5679 \pm 0.0044$ |
| | time-dependent | $\mathbf{0.8345 \pm 0.0090}$ | $1.8720 \pm 0.0520$ | $0.5674 \pm 0.0037$ |
| | constant | $0.6241 \pm 0.0018$ | $1.2079 \pm 0.0241$ | $\mathbf{0.5580 \pm 0.0012}$ |
| Bank Heist | ✗ | $0.7066 \pm 0.0134$ | $1.8289 \pm 0.0330$ | $0.5811 \pm 0.0052$ |
| | time-dependent | $\mathbf{0.9243 \pm 0.0048}$ | $2.8099 \pm 0.0535$ | $0.5991 \pm 0.0080$ |
| | constant | $0.8133 \pm 0.0083$ | $1.9123 \pm 0.0335$ | $\mathbf{0.5464 \pm 0.0025}$ |
| H.E.R.O. | ✗ | $0.2252 \pm 0.0009$ | $0.3712 \pm 0.0122$ | $\mathbf{0.2849 \pm 0.0057}$ |
| | time-dependent | $\mathbf{0.6850 \pm 0.0121}$ | $0.9035 \pm 0.0266$ | $0.3316 \pm 0.0041$ |
| | constant | $0.5827 \pm 0.0102$ | $0.7158 \pm 0.0330$ | $0.3337 \pm 0.0025$ |
| Road Runner | ✗ | $0.2646 \pm 0.0128$ | $1.2432 \pm 0.1125$ | $0.6079 \pm 0.0277$ |
| | time-dependent | $\mathbf{0.7507 \pm 0.0086}$ | $2.9315 \pm 0.0277$ | $0.6419 \pm 0.0012$ |
| | constant | $0.4971 \pm 0.0073$ | $1.6267 \pm 0.1351$ | $\mathbf{0.5652 \pm 0.0078}$ |
| FaceMed | ✗ | $0.2789 \pm 0.0039$ | $0.1329 \pm 0.0035$ | $\mathbf{0.1594 \pm 0.0005}$ |
| | time-dependent | $\mathbf{0.7169 \pm 0.0322}$ | $1.0421 \pm 0.0232$ | $0.2311 \pm 0.0066$ |
| | constant | $0.5897 \pm 0.0227$ | $0.7632 \pm 0.0212$ | $0.4753 \pm 0.0032$ |

The proposed regularization is conceptually and mathematically simple, but requires choosing a set of regularization coefficients $\{\lambda_i\}_{i=1}^{l}$ from a combinatorially large space, making an exhaustive search infeasible in practice. This is not unique to our approach: Real-world sequences often display non-stationary statistical properties that are difficult to model in a data-driven fashion. Nevertheless, we find that our simple coefficient selection protocol leads to significant improvement in calibration, although more sophisticated strategies could well result in further gains.

Finally, we note that since sequential probability estimation as we formulate it—a single model producing calibrated marginal, conditional, and time-to-event estimates—is not clearly addressed by existing methods, our experiments compare variants within the foCus framework. We view the curation of dedicated benchmarks and the development of competing methods as natural future research directions. Other potentially fruitful directions for future research are to perform uncertainty estimation for continuous-valued and spatiotemporal sequences in weather and climate prediction applications—areas in which neural-network simulators are rapidly gaining popularity (Pathak et al., 2022; Kochkov et al., 2024; Subel & Zanna, 2024).

## 8 Conclusion

In this paper, we studied probabilistic sequence prediction, an under-explored task that requires producing calibrated uncertainty estimates for sequences conditioned on high-dimensional inputs, as opposed to simply finding the most likely sequence. Specifically, (1) we proposed a Monte Carlo framework for estimating uncertainty measures for sequences using learned autoregressive simulators, (2) we proposed a suite of evaluation metrics tailored to sequence-derived uncertainty estimates to study their calibration and discrimination, (3) we performed experiments on a new synthetic benchmark dataset for sequence prediction and on real-world data that revealed that maximum likelihood estimation of neural network simulator models is prone to time-dependent miscalibration, and (4) we proposed a simple, time-dependent regularization method for learning autoregressive neural network simulators that leads to well-calibrated uncertainty estimation under the proposed Monte Carlo framework. We hope that this work will inform design choices in uncertainty estimation for sequence prediction models.

## Acknowledgments

This work was supported in part through the NYU IT High Performance Computing resources, services, and staff expertise. W.Z. was supported by the National Institute On Aging of the National Institutes of Health under Award R01AG079175, Award R01AG085617, and NSF grant NRT-1922658. C.F.G. was partially supported by NSF grant DMS 2009752.

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

# Appendix

## Outline

The appendix is organized as follows:

- In Appendix A, we define the metrics we used to evaluate marginal and conditional probability estimation (Appendix A.1) and confidence interval estimation (Appendix A.2).
- In Appendix B, we provide additional information about the datasets utilized in this study.
- In Appendix C, we include additional details regarding the model and hyperparameter search.
- In Appendix D, we include additional results. Appendix D.1 reports results for marginal probability estimation with comprehensive metrics and more reliability diagrams. We also include comparisons with the ground truth probabilities for the synthetic FaceMed dataset. In Appendix D.2, we report results for conditional probability estimation. In Appendix D.3, we report results for confidence interval estimation.
- In Appendix E, we provide a sensitivity analysis of the effect of regularization applied at different entries in a sequence.
- In Appendix F, we present an example illustrating how the probabilities estimated by foCus change when we condition on a certain event.
- In Appendix G, we provide results on applying regularization on a transformer based simulator.
- In Appendix H, we show the evolution of the performance of foCus as we train the simulator with different types of regularization.

## A    Evaluation metrics

### A.1    Marginal and conditional probability

For each class $a \in \{1, \cdots, c\}$ and each entry $i$ of the sequence, we evaluate the estimated probabilities $P(Y_i = a \mid X = x)$ or $P(Y_i = a \mid Y_j = b, X = x)$ using the relevant data (for marginal probabilities, these are all sequences; for conditional probabilities, all sequences such that the $j$th entry equals $b$). The entry-level metrics we propose are defined below. The sequence-level metrics are computed by averaging the entry-level metrics across all entries.

**Macro AUC.**    The Area Under the Curve (AUC) per class is computed separately for each class $a$ at each entry $i$ using a one-vs-all approach. We aggregate all the class AUCs via averaging to obtain the overall macro AUC.

**Brier Score**    The Brier Score (BS) evaluates both calibration and discriminative ability. The Brier Score per class is the mean-squared error between the predicted probabilities and binarized label per class. The entry-level BS is the mean of BS per class, averaged over all the classes.

**Cross Entropy**    The cross-entropy (CE) loss is computed following equation 3.

**Expected Calibration Error.**    We use confidence expected calibration error (ECE) (Guo et al., 2017) to assess calibration. The *confidence* is defined as the predicted probability of the class $a$ with the highest estimated probability. These confidences are grouped into $B$ bins, based on $B$-quantiles. ECE is the mean absolute difference between the accuracy (empirical probability of correct predictions) and the average confidence within each bin. A lower ECE indicates better calibration.

To provide further insight, we also plot reliability diagrams. These diagrams compare the empirical probability (accuracy) with the estimated probability (confidence) in each bin. A well-calibrated model will produce a reliability diagram that is close to the diagonal.

## A.2 Confidence intervals

Let the ground truth time-to-event for the $k$th data point be denoted as $T[k]$, and the estimated confidence interval as $I_\alpha[k] = \left[q_{(1-\alpha)/2}[k], q_{(1+\alpha)/2}[k]\right]$.

**Coverage Probability** The coverage probability measures the proportion of samples where the true time-to-event $T[k]$ lies within the estimated confidence interval $I_\alpha[k]$. This metric reflects how well calibrated the estimated confidence intervals are. Ideally, for a confidence level $\alpha$, the coverage probability should equal $\alpha$.

**Relative Confidence Interval Width** The width of the confidence interval quantifies the uncertainty in the model estimates. A wider confidence interval indicates higher uncertainty. To account for different sequence lengths across datasets, we normalize the confidence interval width by the mean of the true time-to-event averaged over each dataset. The relative confidence interval width is defined as:

$$\frac{\frac{1}{N}\sum_{k=1}^{N}\left(q_{(1+\alpha)/2}[k] - q_{(1-\alpha)/2}[k]\right)}{\frac{1}{N}\sum_{k=1}^{N} T[k]}, \tag{5}$$

where $N$ is the number of data.

**Relative Mean Absolute Error (MAE)** The relative MAE is the mean of the absolute difference between the estimated and ground truth time-to-event, normalized by the mean of the ground truth times-to-event. We estimate the time-to-event by averaging over the time-to-event values $\widehat{T}^{(1)}, ..., \widehat{T}^{(m)}$ corresponding to the $m$ Monte Carlo simulations:

$$\text{Relative MAE} = \frac{\sum_{k=1}^{N}\left|T[k] - \frac{1}{M}\sum_{m=1}^{M}\widehat{T}^{(m)}[k]\right|}{\sum_{k=1}^{N} T[k]}. \tag{6}$$

## B Datasets

**FaceMed** FaceMed is a synthetic dataset based on the UTKFace dataset (Zhang et al., 2017b), which contains face images along with corresponding ages. We simulate health status transitions between three distinct states: 1 for healthy, 2 for ill, and 3 for dead. The simulated health states per each year form a sequence for each patient. The underlying transition probabilities among these health states are determined by the individual's age. The goal of the sequence prediction task is to forecast a patient's health status trajectory using their facial images.

To simulate the dynamics of health status, we use an age-dependent Markov process, where the health status at the $i$th entry, $Y_i$, only depends on the previous state $Y_{i-1}$, for any $i > 1$. The conditional probability between states is given by:

$$\mathrm{P}(Y_i = a \mid Y_{i-1} = b) = p_{b,a}. \tag{7}$$

The transitions among health statuses are illustrated in Figure 5. Every individual is healthy as the initial state. The transition probabilities for the simulation are defined as follows: For individuals younger than 40, the health status never changes (Figure 5 (a)). For those aged 40 to 80, the health status can transition between healthy and ill, with transition probabilities $p_{1,1} = p_{2,2} = 0.9$, $p_{1,2} = p_{2,1} = 0.1$ (Figure 5 (b)). For individuals older than 80, the likelihood of becoming ill increases, with transition probabilities $p_{0,\cdot} = (0.6, 0.4, 0)$; for ill individuals in this age group, there is a chance of death, reflected in the transition probabilities $p_{1,\cdot} = (0.1, 0.7, 0.2)$ (Figure 5 (c)). These probabilities vary as the individual becomes "older" in the simulation, reflecting the increasing risks associated with aging. The average survival time of the simulated sequences is 45.45 years. In the experiments, for individuals who die before 100 years from the beginning, their sequences are padded to cover 100 years. The dataset is split into training, validation, and test sets with 16641, 4738, and 2329 samples, respectively.

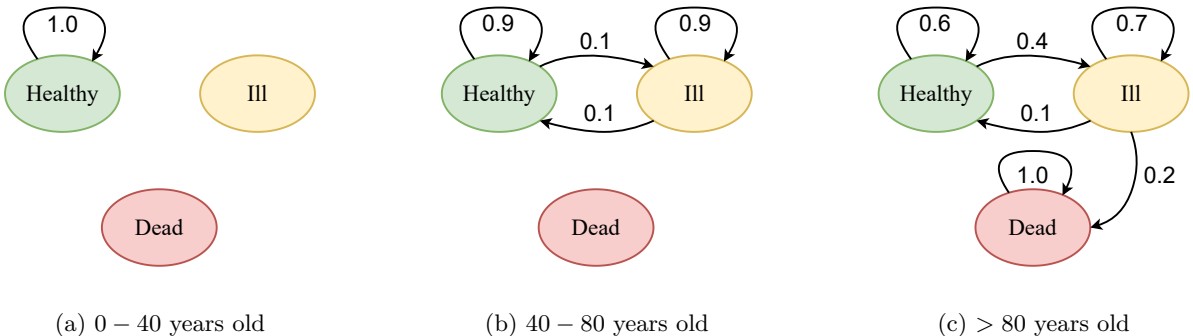

(a) $0 - 40$ years old        (b) $40 - 80$ years old        (c) $> 80$ years old

Figure 5: **Markov process used to simulate health-status transitions in FaceMed.**

Table 4: **Additional information about video game data.** The table summarizes each video game dataset's training set size, validation set size, test set size, average sequence length until the next scoring point, sequence length after padding, and action sampling frequency.

| Game Name | Training | Validation | Testing | Ave. Length (seconds) | Padded Length | Sampling freq. |
|---|---|---|---|---|---|---|
| Seaquest | 104,806 | 17,556 | 17,650 | 1.0071 | 200 | 3 |
| River Raid | 104,252 | 17,584 | 17,592 | 1.4082 | 300 | 2 |
| Bank Heist | 105,765 | 17,553 | 17,548 | 1.7645 | 300 | 2 |
| H.E.R.O. | 98,553 | 16,769 | 16,885 | 1.4586 | 300 | 2 |
| Road Runner | 239,428 | 68,332 | 34,177 | 2.5796 | 300 | 4 |

**Atari games** Our real-data experiments are based on the Atari-HEAD dataset (Zhang et al., 2020b), a large-scale, high-quality imitation learning dataset that captures human actions alongside eye movements and game frames while playing Atari video games. The dataset employs a unique semi-frame-by-frame gameplay format, where the game pauses at each frame until the player performs a keyboard action. This ensures that each frame in the video of game and the corresponding human action are aligned.

In this work, the sequence prediction task aims to predict a player's action trajectory based on a given game frame. Each frame serves as a high-dimensional input $X$. The subsequent actions until the next scoring event are treated as a sequence $Y$. Since actions are recorded at a high frame-by-frame frequency, they often repeat several times before transitioning to a new action, yielding sequences with redundant information. To reduce this redundancy, we sample actions at a constant frequency determined by the number of frames per sequence entry for each game. The corresponding time $t$ in the game at the $i$th entry of the sequence can be recovered from the entry as follows: $t(\text{second}) = \text{sampling freq.}/60(\text{Hz}) \times i$. We pad the sequences with an *end-of-game* value to ensure that all sequences have the same length.

Experiments are conducted on five games from Atari-HEAD: **Seaquest**, **River Raid**, **Bank Heist**, **H.E.R.O.**, and **Road Runner**. They represent a broad category of video games available in Atari-HEAD. We use 70 percent of gameplay as training set, 20 percent as the validation set, and 10 percent as the test set. See Table 4 for more detailed information about the data corresponding to each game.

## C  Technical Details

### C.1  Model Architecture

As illustrated in Figure 2 (a), we employ a neural-network simulator. The simulator consists of a convolutional neural network (CNN) that encodes the input image $x$, and a recurrent neural network (RNN). The CNN encoder is a Resnet-18 (He et al., 2015), which produces an image embedding of dimension 256. The RNN decoder is implemented as a single-layer LSTM with a hidden vector size 256. The image embedding from the CNN encoder is fed to the RNN decoder as the initial hidden vector $h_0$. The RNN iteratively

updates its hidden state $h_i$ based on the previous hidden state $h_{i-1}$ and the preceding input $y_{i-1}$ from the $(i-1)$th entry of the sequence. For each step $i > 1$, the RNN outputs a logit vector $\mathbf{z}(x, y_1, \ldots, y_{i-1}) \in \mathbb{R}^c$ through a linear layer with input $h_i$. The logit vector is normalized with a softmax function and used to estimate the class probabilities of a multinoulli distribution:

$$p_\theta(a \mid x, y_1, \ldots, y_{i-1}) = \frac{\exp(\mathbf{z}(x, y_1, \ldots, y_{i-1})[a])}{\sum_{k=1}^c \exp(\mathbf{z}(x, y_1, \ldots, y_{i-1})[k])}, \tag{8}$$

where $a \in \{1, \ldots, c\}$ and $\mathbf{z}(x, y_1, \ldots, y_{i-1})[k]$ is the $k$th entry of the logit vector.

For the health-status prediction task, the RNN decoder outputs a 3-dimensional logit vector corresponding to the three possible health states ($c = 3$). In the case of Atari games, where each action can belong to one of 19 possible classes, the RNN generates a 19-dimensional logit vector ($c = 19$).

## C.2 Training

As explained in Section 6, the neural network simulator is trained by minimizing the cross-entropy loss between the predicted distribution and the one-hot encoded ground truth for each variable, with an additional regularization term that penalizes the $\ell_2$-norm of each entry in the logit vector.

We train each model for 200 epochs for each scenario, with a batch size of 256, using the Adam optimizer without weight decay. The learning rates are kept constant for each scenario: $1 \times 10^{-5}$ for Seaquest, River Raid, Bank Heist, and Road Runner, and $5 \times 10^{-5}$ for H.E.R.O.

Model selection during training is challenging due to the numerous metrics involved in probability estimation tasks. Figures 13, 14, and 15 show the evolution of different metrics during training. We observe that the models are most discriminative (lower relative MAE and higher AUC) toward the end of training in most scenarios.

## C.3 Inference

Figure 2(b) illustrates how the neural-network simulator is used to obtain a sample sequence $(\hat{y}_1, \ldots, \hat{y}_\ell)$. The input image $x$ is fed into the CNN encoder, producing a hidden vector $h_0$ that is fed into the RNN decoder to then generate the simulated sequence iteratively. At each iteration $i \in \{1, \ldots, \ell\}$, the input of the RNN decoder is the value $\hat{y}_{i-1}$ of the previous entry (except for $i = 1$) and the hidden vector $h_{i-1}$. The outputs are an estimate of the conditional distribution of $Y_i$ given the previous entries and $X = x$, and an updated hidden vector $h_i$. The $i$th entry $\hat{y}_i$ of the sample sequence is sampled from this conditional distribution. Since each entry is drawn randomly from the predicted distribution, the simulator is capable of generating multiple different sequences from the same input image, acting as a simulator. When performing Monte Carlo estimation, we generate $m = 100$ sampled sequences for each input image.

## C.4 Hyperparameter search

The hyperparameters for time-dependent regularization were determined via the following procedure:

1. For $1 \le i \le k_1$ (where $k_1$ is a hyperparameter) we use the sequence-level ECE of marginal probabilities (see Section 4.1) computed over validation set to iteratively select $\lambda_i$, setting $\lambda_j = 0$ for all $j > i$.

2. For $k_1 < i \le k_2$ (where $k_2$ is a hyperparameter) we constrain all the parameters to equal the same constant, $\lambda_i = \lambda_{\text{all}}$, selected also based on the validation ECE.

3. For $i > k_2$ we set $\lambda_i = 0$.

We set $k_1 = 3$. To determine each $\lambda_i$ and $\lambda_{\text{all}}$ we performed a search on the fixed grid $\{0.001, 0.005, 0.01, 0.05\}$ based on the validation ECE for the marginal probability estimation task. For $k_2$ we used the grid $\{1, 11, 21, 51, 101\}$.

Table 5: **Chosen regularization hyperparameters.** This table shows the chosen regularization parameters in all scenarios for both time-dependent and constant regularization.

| Scenarios | Time-dependent $\lambda$'s | | | Constant $\lambda$'s |
|---|---|---|---|---|
| Seaquest | $\lambda_{1:3} = 0.05, 0.01, 0.05$ | $\lambda_{4:200} = 0$ | | $\lambda_{1:200} = 0.001$ |
| River Raid | $\lambda_{1:6} = 0.01$ | $\lambda_{7:300} = 0$ | | $\lambda_{1:300} = 0.001$ |
| Bank Heist | $\lambda_1 = 0.05$ | $\lambda_{2:11} = 0.01$ | $\lambda_{12:300} = 0$ | $\lambda_{1:300} = 0.001$ |
| H.E.R.O. | $\lambda_1 = 0.01$ | $\lambda_{2:6} = 0.005$ | $\lambda_{7:300} = 0$ | $\lambda_{1:300} = 0.001$ |
| Road Runner | $\lambda_1 = 0.01$ | $\lambda_{2:21} = 0.005$ | $\lambda_{22:300} = 0$ | $\lambda_{1:300} = 0.001$ |
| FaceMed | $\lambda_{1:3} = 0.01$ | $\lambda_{4:5} = 0.005$ | $\lambda_{5:50} = 0.001$ | $\lambda_{1:300} = 0.001$ |

For constant regularization, we constrain all the parameters to be the same, $\lambda_i = \lambda_{\mathrm{const}}$. Then $\lambda_{\mathrm{const}}$ we performed a search on the grid $\{0.001, 0.005, 0.01, 0.05\}$, also based on validation ECE. The hyperparameters chosen for both regularization methods are listed in Table 5.

## D Supplementary Experimental Results

### D.1 Marginal Probability

Figure 6 shows plots of the entry-level metrics (ECE, AUC, BS, and CE) for marginal probability estimation, complementing Figure 3. Time-dependent regularization leads to a substantial improvement in calibration, as demonstrated by the significantly lower ECE of the time-regularized model, which has a comparable AUC to the model without regularization. As a result, the cross-entropy and Brier Score metrics are also improved. Constant regularization also improves the probability estimates, but not as much as time-dependent regularization. Figure 7 shows additional reliability diagrams like the ones in Figure 3 and includes a comparison with constant regularization, confirming that time-dependent regularization consistently improves calibration for individual entries.

In Section 6.1, we also compare the estimated marginal probability with underlying data generating distribution on the synthetic FaceMed data. Since the ground truth sequences in FaceMed are generated using an assumed transitional probability model, as detailed in Appendix B, we can analytically compute the *ground truth* marginal probability of $Y_i$ based on the transitional probabilities and the marginal probability of the previous entry, $Y_{i-1}$:

$$\mathrm{P}(Y_i = a \mid X = x) = \sum_{b=1}^{c} \mathrm{P}(Y_{i-1} = b \mid X = x)\,\mathrm{P}(Y_i = a \mid Y_{i-1} = b) \tag{9}$$

Starting from an initial state of "healthy," we compute the ground truth marginal probabilities for each entry in the sequence and compare them against the estimates obtained via foCus. We calculate the root mean square error (RMSE) between the ground truth and the estimated marginal probabilities for each entry, then aggregate these values to derive a sequence-level RMSE by averaging. The unregularized baseline method and constant regularization yield RMSE of $0.1847 \pm 0.0014$ and $0.1754 \pm 0.0028$, respectively, while time-dependent regularization reduces the RMSE to $0.1720 \pm 0.0009$. This result further demonstrates the improved performance achieved by time-dependent regularization.

### D.2 Conditional Proabability

We evaluate the conditional probability estimation, given a fixed event in each scenario. For conditional probability estimation, FaceMed is conditioned on the status of the first year being "Healthy". Video games are conditioned on the first entry being equal to the most frequent first action in the training set. Specifically, Seaquest is conditioned on first action as *NOOP* (no operation), River Raid as *NOOP*, Bank Heist as *Right*, H.E.R.O. as *NOOP*, and Road Runner as *Left*. Table 2 reports the sequence-level metrics for conditional probability estimation. Figure 8 shows entry-level metrics (ECE, AUC, BS, and CE) for conditional prob-

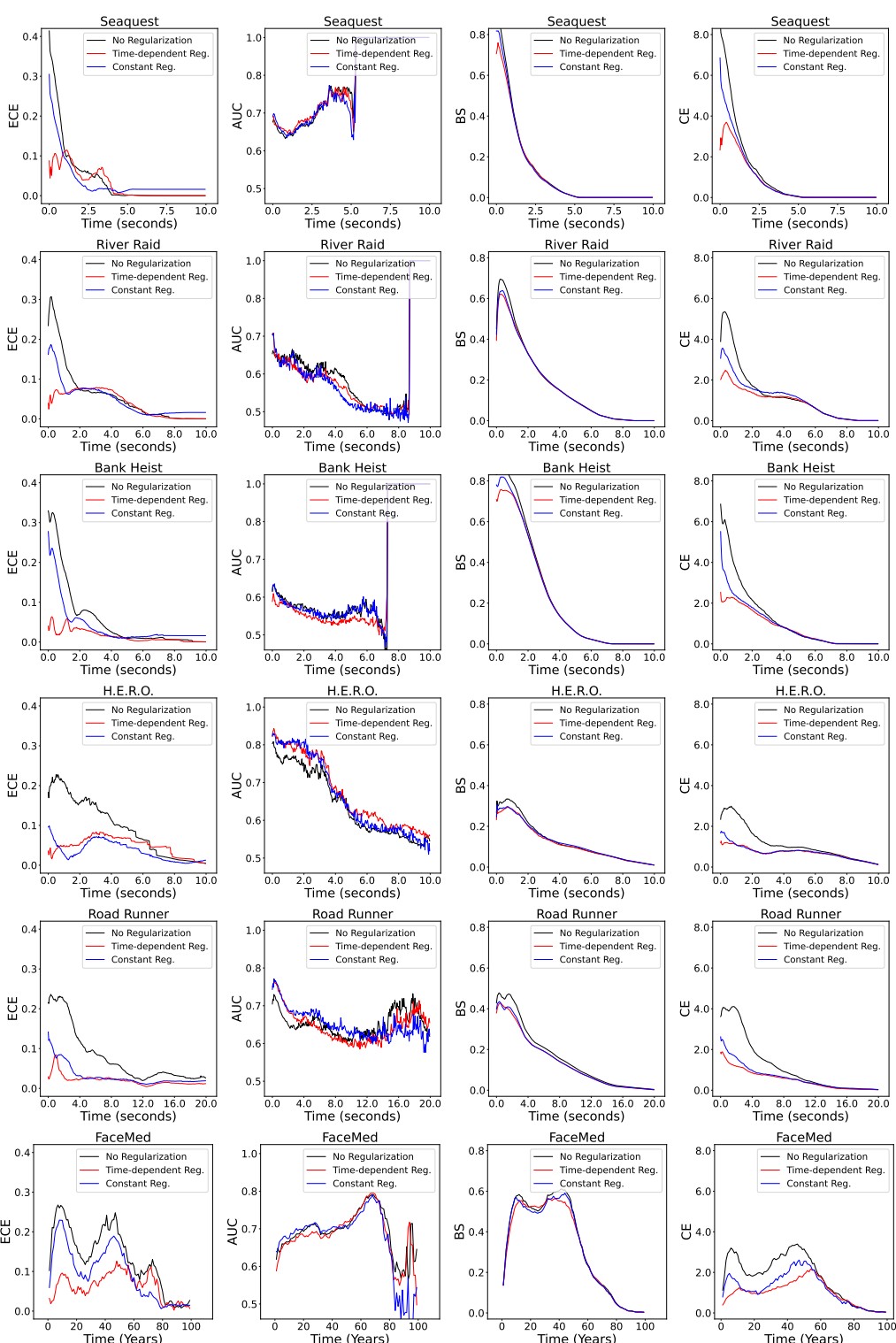

Figure 6: **Entry-wise metrics for marginal probability estimation.** The figure plots the entry-level ECE, AUC, BS, and CE of the proposed foCus framework for estimation of marginal probabilities. It compares versions of foCus: without regularization (black), time-dependent regularization (red), and constant regularization (blue). Both constant regularization and time-dependent regularization improve calibration and overall estimation quality compared to without regularization. Time-dependent regularization's improvement is more significant than constant regularization.

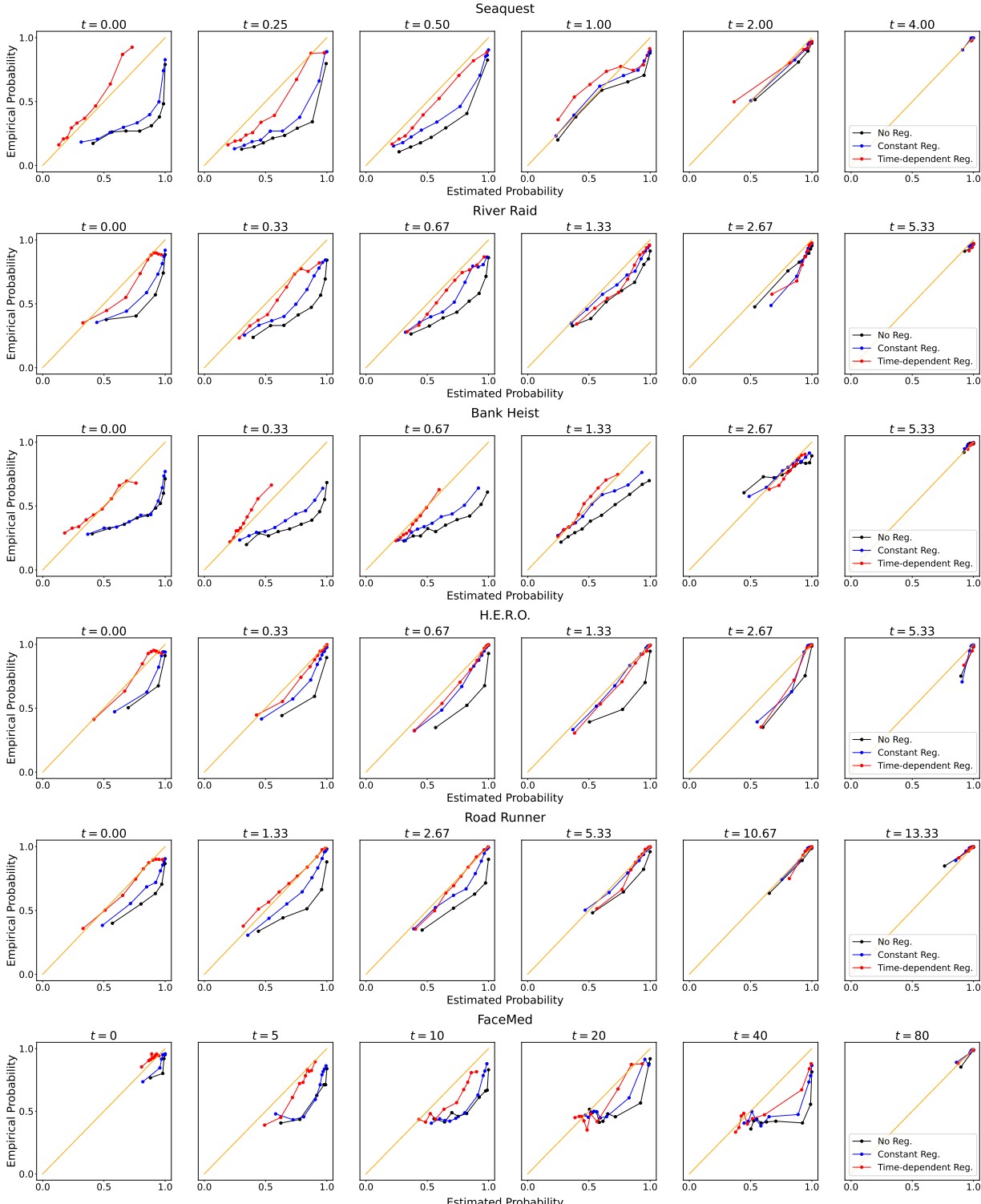

Figure 7: **Entry-wise reliability diagrams for marginal probability estimation.** The figure, supplementing Figure 3 presents reliability diagrams at additional entries in the sequence. Each diagram compares three versions of the proposed foCus framework: without regularization (black), time-dependent regularization (red), and constant regularization (blue). The plots demonstrate that both regularization methods improve calibration at each sequence entry, with time-dependent regularization showing the most substantial improvements.

ability estimation. Similar to the case of marginal probability estimation Appendix D.1, time-dependent regularization improves calibration and the overall quality of probability estimation.

### D.3 Confidence Interval for Time-to-event

Table 3 reports the results for time-to-event confident interval estimation. In this case, we observe a certain trade-off between discriminative performance, quantified by the relative MAE, and calibration, quantified by coverage probabilities. The MAE of no regularization and constant regularization are consistently lower than those of time-dependent regularization, but the coverage probabilities of time-dependent regularization are a lot closer to 90% (between 69% and 92%, compared to at most 70% for the other two methods).

Figure 9, complementing Figure 4, presents heatmaps of the confidence interval widths for different ground-truth time-to-event (upper panel of each subplot) for River Raid, H.E.R.O., Road Runner, and Bank Heist, as well as a histogram showing the fraction of intervals containing the ground-truth (lower panel of each subplot). Unregularized foCus produces very narrow confidence intervals with very poor coverage, whereas time-dependent regularization yields intervals that tend to be larger when the true time-to-event becomes greater (and hence generally more uncertain), achieving much better coverage. These findings further support that time-dependent regularization performs better in confidence interval estimation.

## E Regularization Sensitivity Analysis

In this section, we explore how regularization at a single entry affects model performance. Using the Seaquest game, we train four versions of foCus, each applying regularization to a different entry in the sequence, corresponding to 0, 3, 6, and 9 seconds. As shown in Tables 6 and 7, the model with regularization applied at the beginning (0 seconds) exhibits the most significant improvement in calibration, both for marginal probability estimation and time-to-event prediction confidence intervals. This suggests that applying regularization early in the sequence is crucial for maintaining proper calibration throughout the whole sequence. These results also validate our heuristic procedure to implement time-dependent regularization.

Table 6: **Marginal probability estimation performance sensitivity analysis on Seaquest.** We compare versions of foCus where regularization is applied to a single random variable which is 0, 3, 6, and 9 second(s) after the start of the sequence. Results are presented as mean $\pm$ standard error from three independent model realizations. Best performance is achieved when applying regularization to the random variable immediately after the sequence starts.

| Reg. Time | ECE ($\downarrow$) | AUC ($\uparrow$) | CE ($\downarrow$) | BS ($\downarrow$) |
|---|---|---|---|---|
| 0 second | **0.0290 $\pm$ 0.0008** | 0.8748 $\pm$ 0.0033 | **0.8073 $\pm$ 0.0120** | **0.1164 $\pm$ 0.0008** |
| 3 seconds | 0.0424 $\pm$ 0.0019 | 0.8705 $\pm$ 0.0027 | 1.0227 $\pm$ 0.0150 | 0.1227 $\pm$ 0.0011 |
| 6 seconds | 0.0424 $\pm$ 0.0016 | 0.8783 $\pm$ 0.0024 | 0.9599 $\pm$ 0.0208 | 0.1211 $\pm$ 0.0009 |
| 9 seconds | 0.0409 $\pm$ 0.0020 | **0.8784 $\pm$ 0.0023** | 0.9450 $\pm$ 0.0065 | 0.1212 $\pm$ 0.0013 |

Table 7: **Time-to-event prediction confidence interval performance sensitivity analysis on Seaquest.** The table presents metrics on time-to-event prediction confidence intervals from the same experiments as in Table 6. Regularizing the random variable immediately after the sequence starts significantly improves 90% CI ($I_{0.9}$) coverage.

| Reg. Time | Coverage Prob. of $I_{0.9}$ | Relative Width of $I_{0.9}$ | Relative MAE ($\downarrow$) |
|---|---|---|---|
| 0 second | **0.4972 $\pm$ 0.0208** | 1.6983 $\pm$ 0.0497 | 0.5162 $\pm$ 0.0125 |
| 3 seconds | 0.3577 $\pm$ 0.0171 | 0.7056 $\pm$ 0.0320 | **0.4466 $\pm$ 0.0036** |
| 6 seconds | 0.3556 $\pm$ 0.0099 | 0.7338 $\pm$ 0.0126 | 0.4495 $\pm$ 0.0070 |
| 9 seconds | 0.3578 $\pm$ 0.0016 | 0.7421 $\pm$ 0.0308 | 0.4522 $\pm$ 0.0138 |

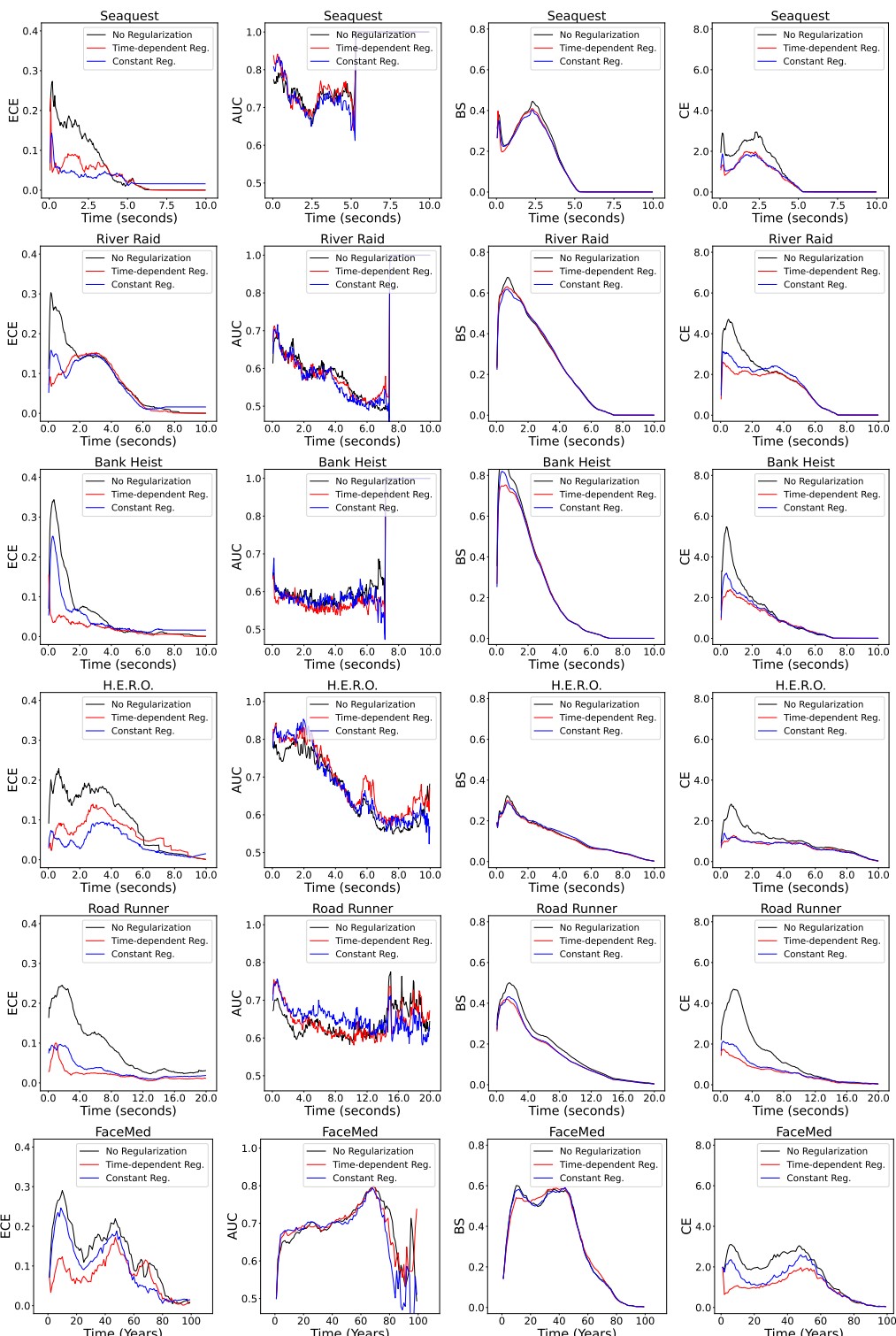

Figure 8: **Entry-wise metrics for conditional probability estimation.** The figure plots the entry-level ECE, AUC, BS, and CE of the proposed foCus framework for estimation of conditional probabilities. It compares versions of foCus: without regularization (black), time-dependent regularization (red), and constant regularization (blue). Both constant regularization and time-dependent regularization improve calibration and overall estimation quality compared to without regularization. Time-dependent regularization's improvement is more significant than constant regularization.

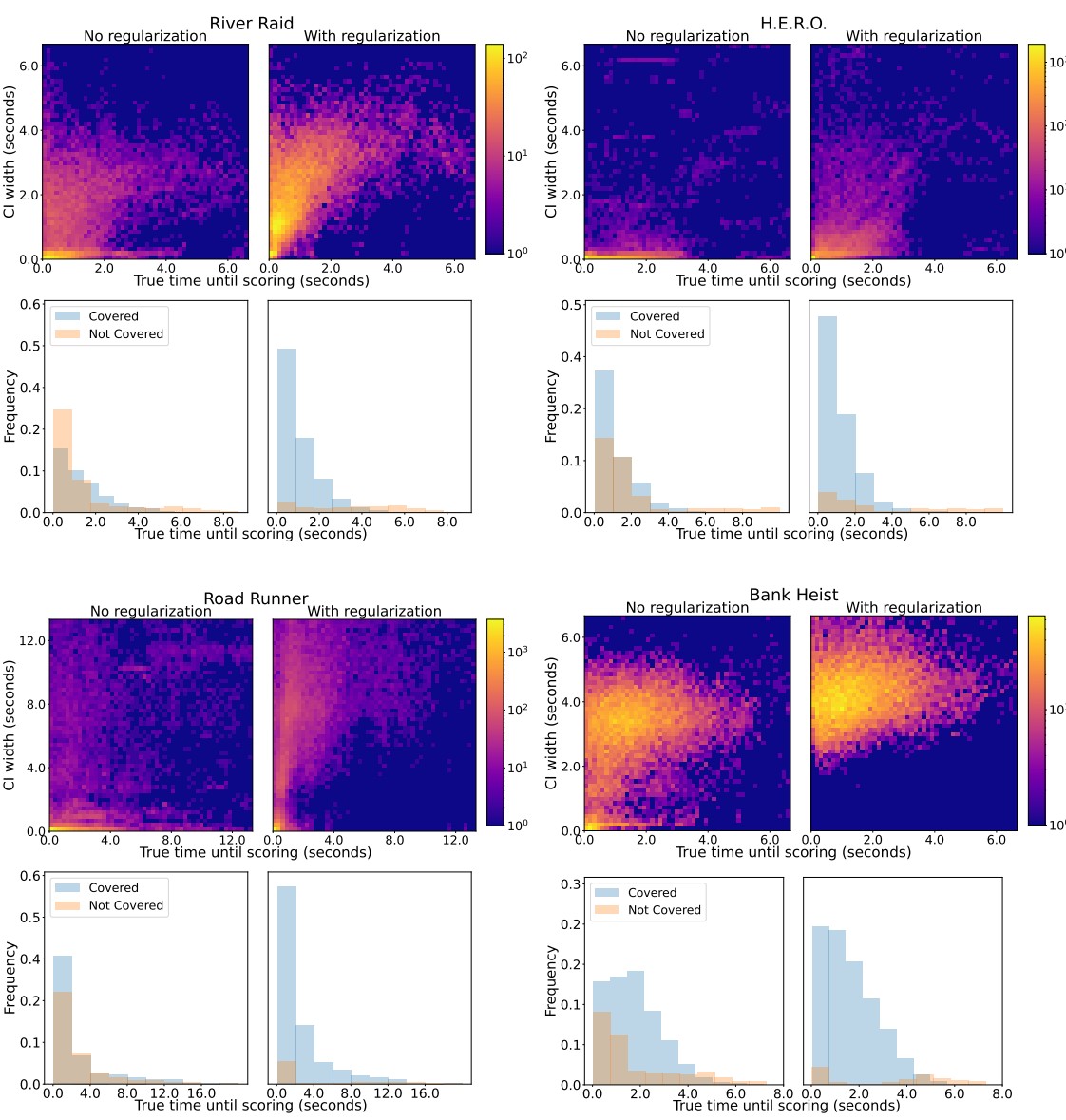

Figure 9: **Confidence intervals for time-to-event prediction and coverage probability.** The upper panel of each subplot shows heatmaps of the length of 0.9 confidence intervals for time-to-event prediction using the proposed foCus framework without and with time-dependent regularization. The histograms below show the frequency of intervals containing the true times, as a function of the true times. Unregularized foCus produces short intervals with poor coverage, whereas regularization yields intervals that tend to be larger when the ground-truth times are larger, and are much better calibrated. The subplots correspond to the River Raid, H.E.R.O., Road Runner, and Bank Heist datasets.

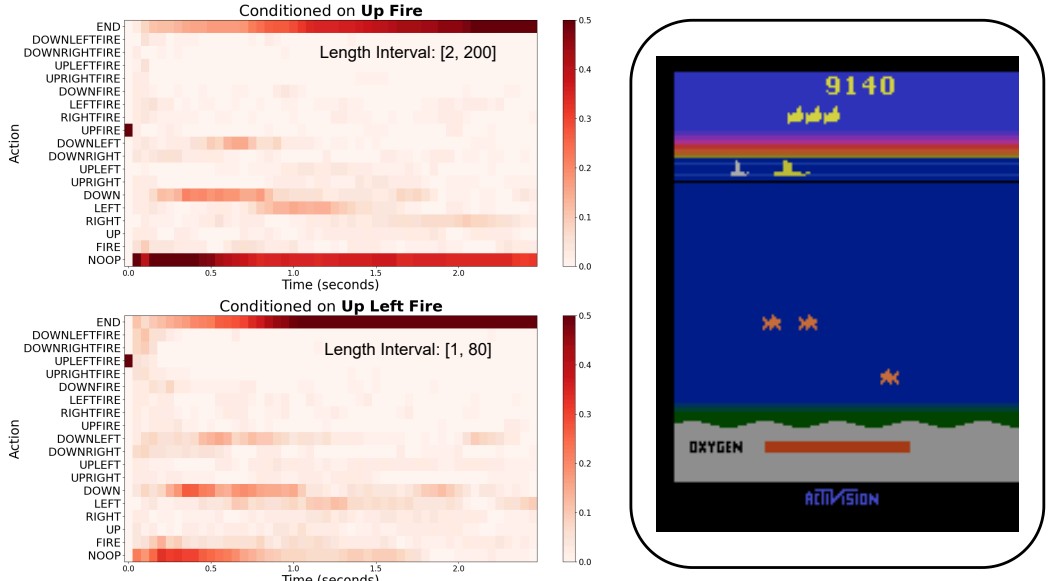

Figure 10: **Action distribution conditioned on different first actions.** The figure shows empirical action distribution for a Seaquest game frame (shown on the right) given first condition actions: Up Fire and action Up Left Fire.

## F    Conditional Probability Estimation

In this section, we illustrate the conditional probability estimation task by showcasing how early actions substantially influence probability predictions at later stages, as shown in Figure 10.

We analyze the probabilities estimated from a frame of the game Seaquest (right panel of Figure 10) by foCus using time-dependent regularization. We estimate the conditional probabilities given two different first-step actions: *Up Fire* and *Up Left Fire*. The left panel of Figure 10 shows the resulting conditional probability estimation. When the first action is *Up Left Fire* instead of *Up Fire*, the frequency of *NOOP* (no operation) decreases significantly, accelerating the player's progress toward scoring. This makes intuitive sense, as *Up Left Fire* moves the green submarine closer to the enemy blue submarine, allowing the torpedo to reach its target more quickly. This analysis illustrates that the model effectively learns meaningful conditional probability estimates.

## G    Experiments with Transformer

In this section, we verify that the regularizer is architecture-agnostic by replacing the RNN simulator with a Transformer decoder. We replaced the RNN component within the original model with a Transformer decoder; and applied this new model architecture to Seaquest and FaceMed. Similar to the RNN based model, the Transformer based model also significantly benefits from the step-wise regularization. Figure 11 shows marginal probability calibration improves with regularization applied. Regularization can also help improve time-to-event calibration for the Transformer based model as in Figure 12.

## H    Supplementary Figures

This section displays the learning curves for various sequence-level metrics on the test set at different epochs during training. All models were trained for 200 epochs, and the metrics were calculated separately for time-to-event prediction confidence intervals Figure 13, marginal probability estimation Figure 14, and con-

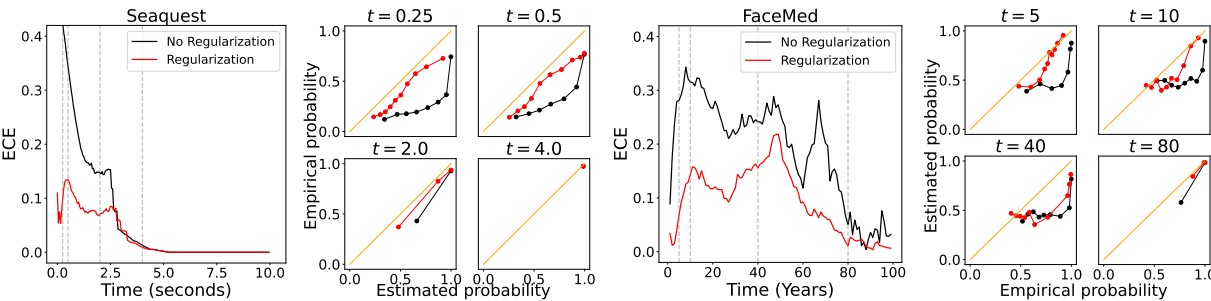

Figure 11: **Entry-wise calibration error and reliability diagrams for marginal probability estimation.** Similar to Figure 3, except the RNN component within the original model is replaced with a Transformer decoder. The results remain consistent, indicating that the findings hold across different sequence model architectures.

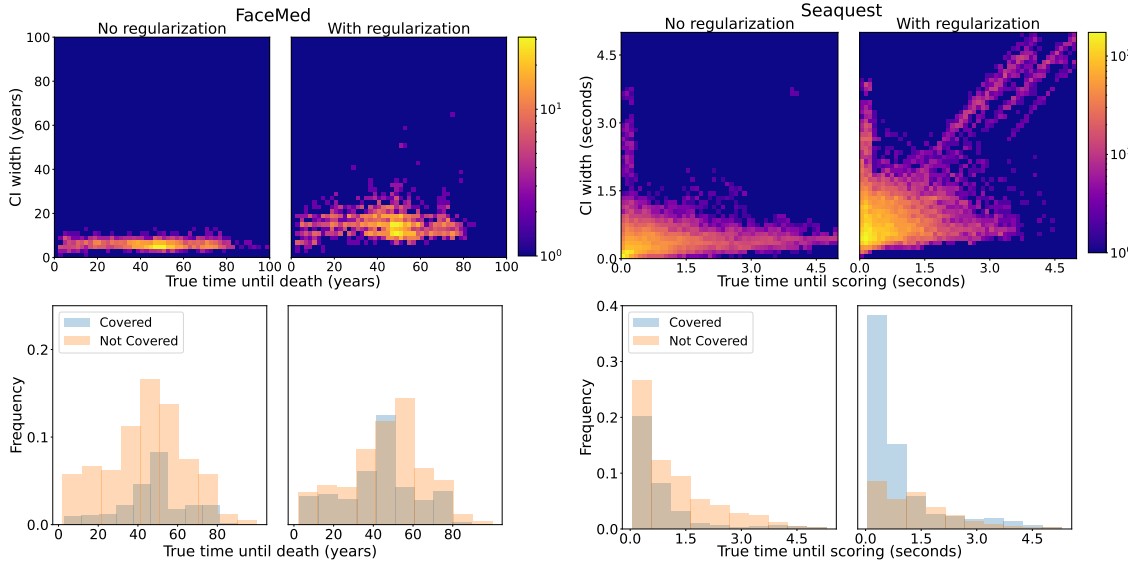

Figure 12: **Confidence intervals for time-to-event prediction and coverage probability.** Similar to Figure 4 but the RNN component within the original model is replaced with a Transformer decoder.

ditional probability estimation Figure 15. These curves highlight the trade-offs between discriminability and calibration over the course of training: lower relative MAE is associated with lower coverage probabilities, and higher AUC tends to come with higher ECE. Despite such trade-offs, the figures demonstrate that among the three foCus variants, the model with time-dependent regularization strikes the best balance, maintaining calibration while improving discriminability throughout training.

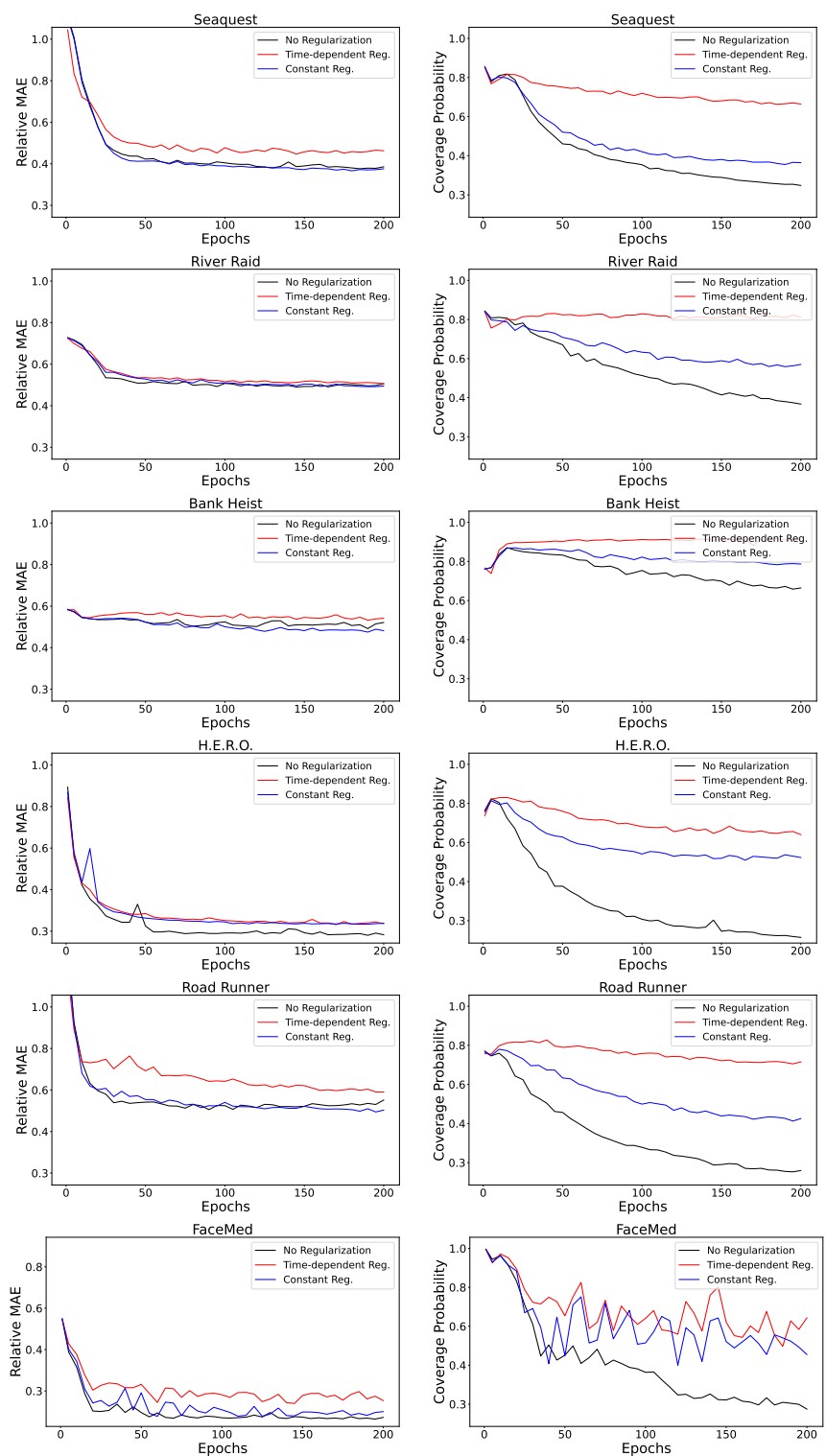

Figure 13: **Time evolution of metrics for time-to-event confident intervals.** It plots how coverage probability and relative MAE of $I_{0.9}$ evolve along training epochs for three versions of foCus: without regularization (black), time-dependent regularization (red), and constant regularization (blue). The time-dependent regularization model achieves significantly better coverage probability.

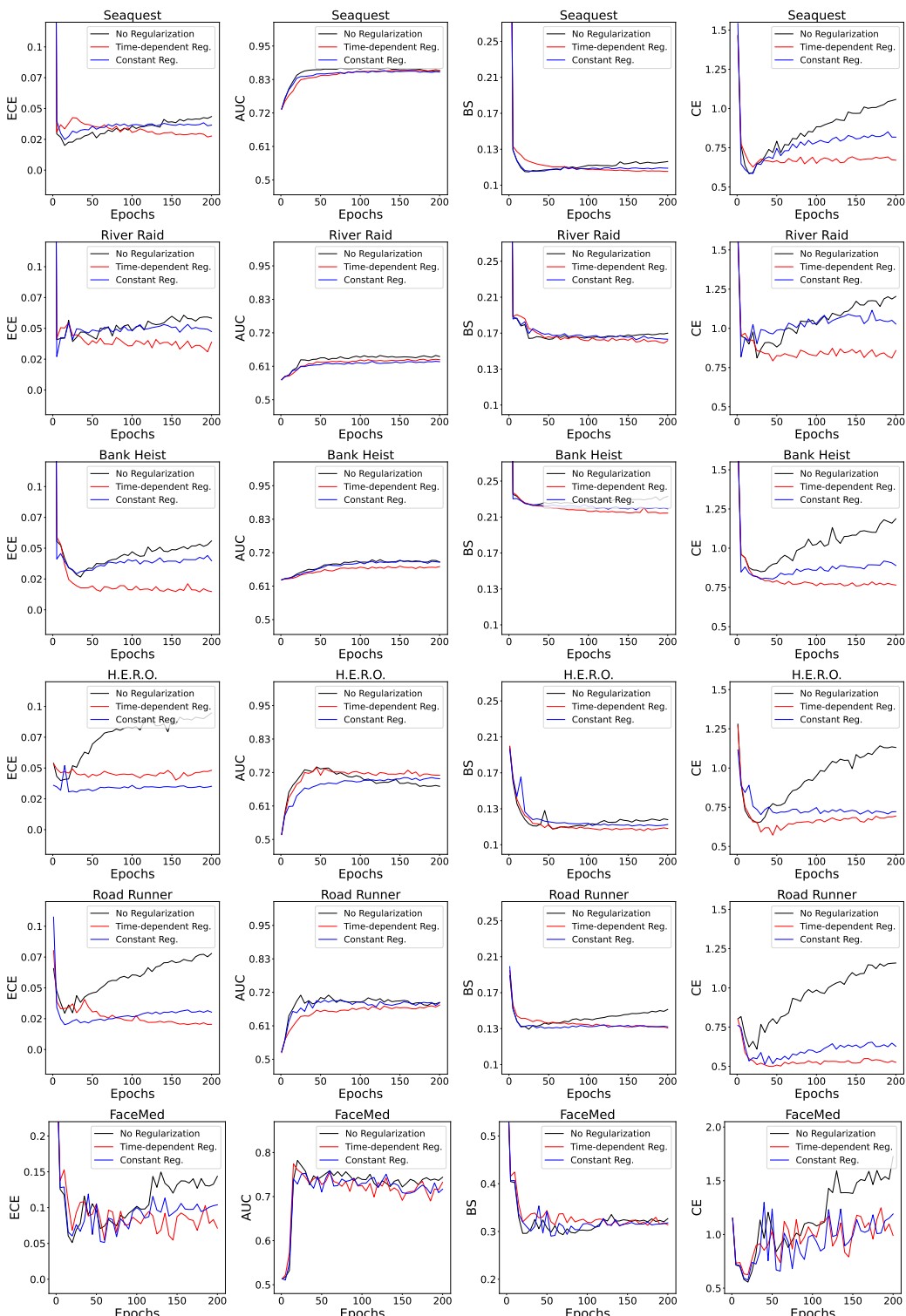

Figure 14: **Learning curves of sequence-level metrics for marginal probability estimation.** It plots how ECE, AUC, BS, and CE evolve along training epochs for three versions of foCus: without regularization (black), time-dependent regularization (red), and constant regularization (blue). The discriminability improves as the calibration decays. The time-dependent regularization model is able to keep calibrated while improving the discriminability.

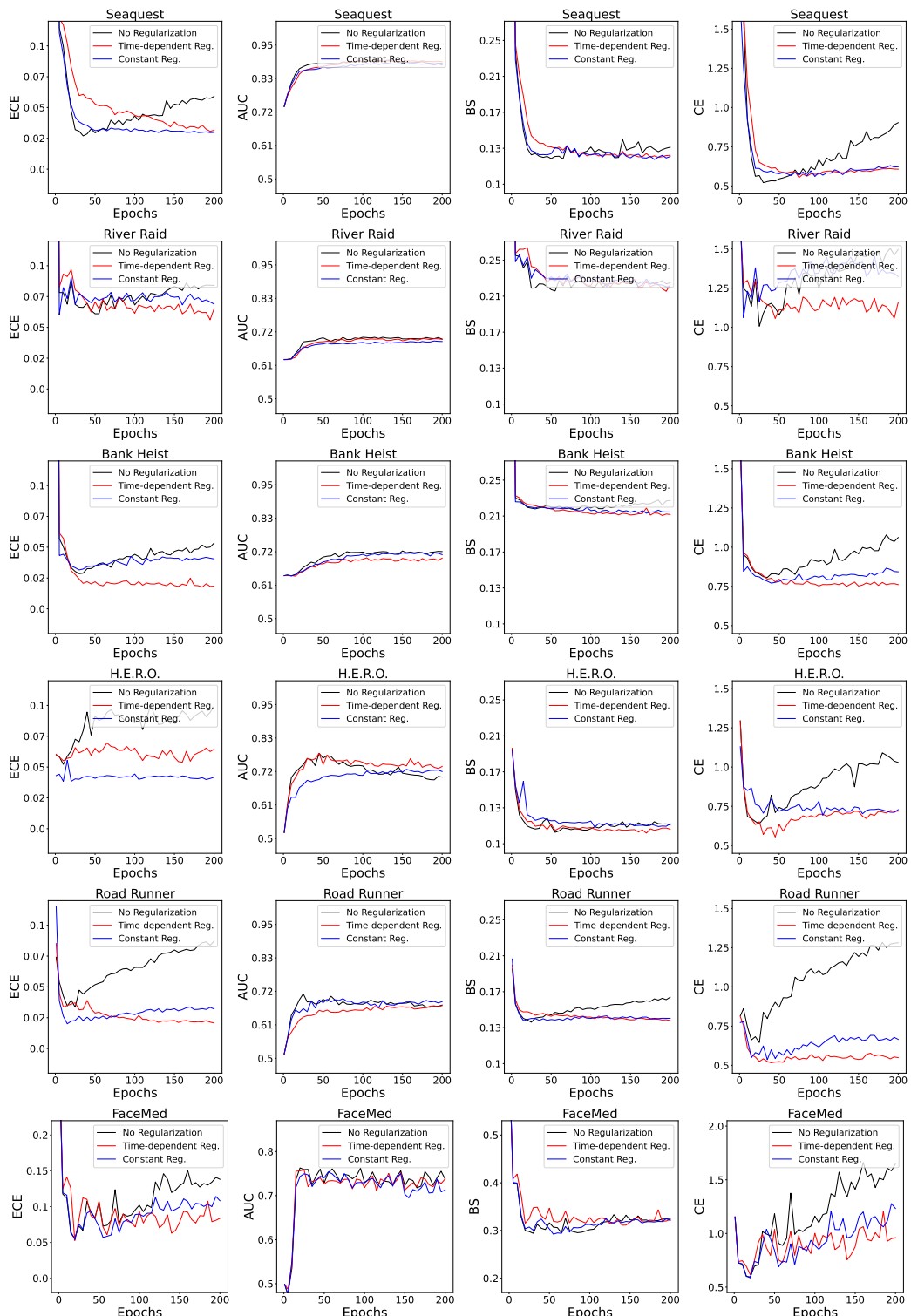

Figure 15: **Learning curves of sequence-level metrics for conditional probability estimation.** A similar pattern in Figure 14 is observed for conditional probability estimation from how ECE, AUC, BS, and CE evolve along training epochs for three versions of foCus.

