# OpenReview forum: "A Monte Carlo Framework for Calibrated Uncertainty Estimation in Sequence Prediction"
_TMLR — Accepted by TMLR_

### Review · Reviewer_tWuc · 2026-03-06

**Summary Of Contributions:**

In many real problems, practitioners care about predicting a sequence given a single input. This work focuses on three types of queries when predicting a discrete sequence: marginal probability of one entry, conditional probability of one entry on another entry, and confidence interval of a time-to-event prediction. Those queries are answered by Monte Carlo simulation after training and simulating an RNN model. However, naive methods do not provide calibration guarantees for the prediction. This work then proposes to add a regularization over the prediction logits at each time-step during training. The resulting model produces better-calibrated predictions for the three queries compared to naive RNNs.

## Strengths
- The idea is clearly explained and the results are easy to interpret.
- The time-dependent regularization makes sense for the problem and the methods are well-supported by empirical evidence.

## Weaknesses
- Using Monte Carlo sampling to answer distributional queries is a standard approach. I am not convinced that it is a contribution of this work.
- The framework is limited to discrete sequences so the regularization has a specific form. I think a more general formalization of the methods will be better appreciated.
- The framework is also restricted to only one input $X$ in the beginning while in many cases we have a sequence of inputs.
- There is not explanation why adding regularization improves the calibration. Many existing works in this area (calibration) stems from probabilistic forecasting (e.g., the cited Kuleshov and Liang, 2015). This work will benefit from a better exposition in this direction.

**Audience:**

Yes

**Audience Explanation:**

The probabilistic forecasting community may appreciate the efforts of increasing calibration with regularization. There has been works in improving calibration by incorporating scoring rules during or after training. The finding that different stages of prediction require different regularization strengths is interesting.

**Broader Impact Concerns:**

I don't see broader impact concerns of this work.

**Claims And Evidence:**

No

**Claims Explanation:**

The main claim in the submission is that adding regularization helps calibration, which are well supported by the experiments. However, the theoretical connections from regularization to calibration is weak. I also don't see why choosing hyperparameters based on the validation ECE for the marginal probability estimation task helps other tasks.

The work claims to propose a framework, but the actual scope is limited. See the weakness section.

**Requested Changes:**

My main request is to make the framework more general: Monte Carlo works in the continuous space, and RNNs can have time-dependent inputs. Moreover, Monte Carlo estimation could be also generalized to expectation of functions which is more general than the three tasks.

I will also appreciate a detailed explanation of the connection between regularization and calibration.

---

> ### Author Response · Authors · 2026-03-25
>
> Q: Using Monte Carlo sampling to answer distributional queries is a standard approach. I am not convinced that it is a contribution of this work.
>
> A: Sequence generation is a well studied topic, but in the literature the probability estimation associated with generated sequences is often overlooked. In this paper, we are bringing people’s attention to this under-explored task. Thus the main contribution of this paper is not building a new approach. But it proposes an under-explored task; designs a suite of metrics to evaluate the performance on this new task; and we propose a simple fix with regularization to improve calibration over the generated sequences. That said, we would be very happy to include any references exploring similar uses of Monte Carlo sampling suggested by the reviewer.
>
> Q: The framework is limited to discrete sequences so the regularization has a specific form. I think a more general formalization of the methods will be better appreciated.
>
> A: Sequences are typically sampled at a discrete set of points, so we believe that the provided formalization is well aligned with practice. However we find this comment intriguing, but are unsure about what is meant with a “general formalization”. If the reviewer can elaborate, we would be happy to consider it.
>
> Q: The framework is also restricted to only one input X in the beginning while in many cases we have a sequence of inputs.
>
> A: The framework can be extended to the case with multiple inputs by embedding the sequence of inputs into a latent vector that can then be provided as the input X. We will mention this in the revised version of the paper.
>
> Q: There is not explanation why adding regularization improves the calibration. Many existing works in this area (calibration) stems from probabilistic forecasting (e.g., the cited Kuleshov and Liang, 2015). This work will benefit from a better exposition in this direction.
>
> A: We thank the reviewer for this valuable suggestion. We agree that a deeper explanation connecting our regularization approach to calibration theory would strengthen the paper. Below we provide this exposition, which we incorporate into the revised manuscript. Following Kuleshov & Liang (2015), calibration requires that predicted probabilities match empirical probability when pooling across events. Neural networks trained via maximum likelihood estimation (MLE)systematically violate this property due to overfitting and memorization: MLE minimizes cross-entropy against one-hot labels, which incentivizes the model to push predicted probabilities toward 0 or 1. In overparameterized networks, this leads to large logit magnitudes that produce overly sharp, low-entropy predictive distributions—even when the true conditional distributions are more uncertain (Guo et al., 2017). Our L2 regularization on logits directly counteracts this mechanism by constraining logit magnitudes. Since extreme probabilities near 0 or 1 require large L2-norm of logits, we prevent the model from producing overconfident predictions by penalizing the logit norm. This mechanism can be explained theoretically through the function-space regularization framework of Rudner et al. (2023): the L2 regularization corresponds to imposing a Gaussian prior centered at zero in logit space, encoding the belief that extreme probabilities require strong evidence. Under this prior, maximum a posteriori estimation (MAP) estimation shrinks predicted probabilities toward higher-entropy distributions when evidence is insufficient. We have expanded Section 6 to include this exposition, explicitly connecting our approach to Kuleshov & Liang (2015) and Rudner et al. (2023).
>
> Q: I also don't see why choosing hyperparameters based on the validation ECE for the marginal probability estimation task helps other tasks.
>
> A: Tasks proposed in this paper intend to measure the performance of estimating joint sequence distribution. Here we choose hyperparameters based on marginal ECE averaged over all sequential steps. In other words, we are hoping the chosen model can estimate each marginal probability correctly. That is the necessary condition for the model to be good at estimating the joint distribution, thus having good performance on the other tasks.

---

> > ### Author Response · Authors · 2026-03-25
> >
> > Q: My main request is to make the framework more general: Monte Carlo works in the continuous space, and RNNs can have time-dependent inputs. Moreover, Monte Carlo estimation could be also generalized to expectation of functions which is more general than the three tasks.
> >
> > A: We thank the reviewer for these insightful suggestions regarding the generalizability of the foCus framework. We agree that the underlying MC + autoregressive approach enables extension to more complex settings: modeling continuous distributions, incorporating time-dependent inputs, and estimating expectation of general functions on sequence space. We would be happy to address these possibilities in the Conclusion/Future Work section explaining how the foCus framework serves as a foundation for these more broad setups. However, here we chose to focus this paper on the discrete case as this setting is already high-dimensional and representative of many real-world safety-critical applications. We believe that a thorough study in this fundamental case is a necessary first step before extending the framework to more complicated settings.

---

### Review · Reviewer_BzgE · 2026-03-10

**Summary Of Contributions:**

In this paper, the authors proposed a Monte Carlo framework to estimate probabilities and confidence intervals associated with sequences. The framework uses a Monte Carlo simulator, implemented as an autoregressively trained neural network, to sample sequences conditioned on an image input. Furthermore, authors also propose a time-dependent regularization. Authors conduct experiments on synthetic and real data show that the framework produces accurate discriminative predictions.

**Audience:**

Yes

**Audience Explanation:**

The topic is important and the task is reasonable.

**Claims And Evidence:**

Yes

**Claims Explanation:**

The authors have conduct extensive experiments. And the authors write lots of theory analysis.

**Requested Changes:**

1. I would like to know if the transformers framework work in this method.

2. Are there any new datasets? I concern about the current test set have been overfitted.

3. I recommend the authors to compare the proposed method with newer baseline methods.

4. After a quick look, I find nearly all the references are earlier than 2025.

---

> ### Author Response · Authors · 2026-03-25
>
> Q: I would like to know if the transformers framework work in this method.
>
> A: Yes, the foCus framework is architecture-agnostic and works with Transformers. We have included experiments with Transformer-based simulators in Appendix G, which demonstrate comparable results to RNN-based models. The time-dependent regularization approach applies equally to both architectures since it operates on the output logits rather than the internal architecture.
>
>
> Q: Are there any new datasets? I concern about the current test set have been overfitted.
>
> A: We appreciate the reviewer's concern about overfitting. We followed standard evaluation protocols with strict train/validation/test splits (7:2:1 ratio) and report results averaged over three independent model realizations with standard errors (Tables 1, 4, 5). Hyperparameters were selected using validation ECE, and test sets were held out until final evaluation. For FaceMed, we additionally validate against known ground-truth marginal probabilities derived from the data-generating process, providing an independent check on model quality. Regarding new datasets: since sequential probability estimation is a newly proposed task, dedicated benchmarks do not yet exist. We believe curating such benchmarks would be valuable for the community and represents a promising direction for future work.
>
>
> Q: I recommend the authors to compare the proposed method with newer baseline methods.
>
> A: Sequence generation is a well studied topic, but the probability estimation associated with generated sequences is often overlooked. In this paper, we are bringing people’s attention to this under-explored task. We have found it hard to identify comparable baselines. If you have any suggestions, we would be very happy to include them.
>
>
> Q: After a quick look, I find nearly all the references are earlier than 2025.
>
> A: We thank the reviewer for this observation. This paper has been under review at TMLR for approximately six months, which accounts for the reference timeline. We would be happy to include any additional references that the reviewer finds appropriate.

---

### Review · Reviewer_LMks · 2026-03-18

**Summary Of Contributions:**

This paper proposed to use model probability of sequential predictions $P(Y_i=y_i|X=x, Y_{i-1}=y_{i-1},...,Y_1=y_1)$ with a Recurrent Neural Network and proposed to use Monte-Carlo method to marginalize the probability into $P(Y_i=y_i|X=x, Y_j=y_j)$,  $P(Y_i=y_i|X=x)$, .etc. Additionally, a regularization method is proposed to improve the calibration performance of the proposed method.

**Audience:**

Yes

**Audience Explanation:**

Model calibration in sequential prediction could be important for safety critical problems like health prediction.

**Claims And Evidence:**

No

**Claims Explanation:**

Theoretically speaking, the proposed method is straightforward to understand. Actually, I am surprised that the idea of using Monte-Carlo method and RNN to model sequential prediction probability have not been proposed before.

Empirically speaking, the proposed method is more or less isolated, without comparing with any other previous works. Even though the authors emphasize that this paper focus on the calibration performance of the model compared with previous sequence generation works, this does not means there is no need to compare with those works. Therefore I cannot see a strong evidence demonstrating the effectiveness of the model.

**Requested Changes:**

1. Compare with previous works, at least two.
2. Increase the amount of contribution, current version is not very sufficient (in my opinion, MC is really straightforward, only the regularization could count)

---

> ### Author Response · Authors · 2026-03-25
>
> Q: Empirically speaking, the proposed method is more or less isolated, without comparing with any other previous works. Even though the authors emphasize that this paper focus on the calibration performance of the model compared with previous sequence generation works, this does not means there is no need to compare with those works. Therefore I cannot see a strong evidence demonstrating the effectiveness of the model.
>
> A: Thanks for your valuable comments. As we discussed in the paper sequence generation is a very well studied topic with a lot of applications LLM for example, however, those works rarely focus on the probability estimation associated with the generated sequences. To address this gap, this paper initiates a combination of MC with mainstream autoregressive sequence generation to study sequential probability estimation. With this regard, our paper proposes a new task and we make the very first attempt to solve this task using the most intuitive method: MC + autoregressive sequence generation. Then we design a new suite of rigorous metrics to evaluate the performance on sequential probability estimation. Targeting the demonstrated miscalibration, we propose a simple fix with regularization to improve calibration over the generated sequences. Given the nature of this paper: solving a new under-explored task, we have found it hard to identify comparable baselines. If you have any suggestions, we would be very happy to include them.
>
>
> Q: Increase the amount of contribution, current version is not very sufficient (in my opinion, MC is really straightforward, only the regularization could count)
>
> A: As we discussed above, the main contribution of this paper is not building a new method. But it proposes a new task; designs a suite of metrics to evaluate the performance on this new task; and proposes a simple fix to a major miscalibration issue encountered by a reasonable approach on this new task.

---

### Decision · Action_Editor_1xJ2 · 2026-05-13

**Recommendation:** Accept with minor revision

**Additional Comments:**

The paper studies sequential probability estimation and calibration for autoregressive sequence generation using Monte Carlo simulation and time-dependent regularization. The authors focus on evaluation metrics for sequential probability estimation and propose a regularization strategy to improve calibration across generated sequences. This submission was reviewed by three experts, who provided constructive suggestions and valuable comments throughout the review process.

* Reviewer BzgE found the topic interesting and considered the empirical evidence generally sufficient. The reviewer requested stronger baselines, Transformer-based experiments, and clarification regarding dataset overfitting, which were addressed in the rebuttal through additional experiments and clarification of the evaluation protocol.
* Reviewer tWuc appreciated the clarity of the paper and found the regularization strategy empirically effective, but raised concerns about the limited scope of the framework and the connection between regularization and calibration. The rebuttal provided additional discussion connecting the regularization to calibration theory and clarified the motivation for focusing on the discrete setting.
* Reviewer LMks remained unconvinced by the overall contribution strength, noting that the method is mainly compared with variants of itself and lacks stronger external baselines. The rebuttal clarified that the main contribution of the paper is the formulation of an under-explored task together with evaluation metrics and a calibration-oriented regularization strategy, although concerns about empirical breadth still remain. Please make this point clearer in the revision.

Based on the discussion above, I recommend acceptance. While the scope is somewhat limited and the methodological novelty is moderate, the paper studies a meaningful problem and the proposed regularization consistently improves calibration performance.

**Audience:**

Yes

**Audience Explanation:**

Researchers working on uncertainty estimation and calibration may find the problem setting and empirical findings relevant, particularly for safety-critical applications involving sequential predictions.

**Claims And Evidence:**

Yes

**Claims Explanation:**

The paper provides consistent empirical evidence that the proposed regularization improves calibration across multiple sequential probability estimation tasks. While the overall scope is somewhat limited and the comparisons mainly involve variants of the proposed framework, the experiments and additional clarification in the rebuttal are generally sufficient to support the main claims of the paper.